# Aerosol radiative forcings induced by the substantial changes in anthropogenic emissions over China during 2008–2016

Mingxu Liu and Hitoshi Matsui

Graduate School of Environmental Studies, Nagoya University, Nagoya, Japan, 464-8601

Correspondence: Hitoshi Matsui (matsui@nagoya-u.jp)

**Abstract.** Anthropogenic emissions in China play an important role in altering the global radiation budget. In the recent decade, the strong clean-air policies in China have resulted in substantial reductions of anthropogenic emissions of sulfur dioxide ($SO_2$) and primary particulate matter, and air quality over China has consequently improved. However, the resultant aerosol radiative forcings have been poorly understood. In this study, we used an advanced global climate model integrated with the latest

localized emission inventory to quantify the aerosol radiative forcings by the changes of anthropogenic emissions in China between 2008 and 2016. By comparing with multiple observation datasets, our simulations reproduced the considerable reductions of sulfate and black carbon (BC) mass loadings reasonably well over eastern China (the key region subject to stringent emission controls) during the period and accordingly showed a clear decline in both aerosol optical depth and absorption aerosol optical depth. The results revealed a regional annual mean positive direct radiative forcing (DRF) of +0.29

W m$^{-2}$ at the top of the atmosphere (TOA) due to the reduction of $SO_2$ emissions. This positive aerosol radiative forcing comprised of diminished sulfate scattering (+0.58 W m$^{-2}$) and enhanced nitrate radiative effects (–0.29 W m$^{-2}$), and could be completely offset by the concurrent reduction of BC emissions that induced a negative BC DRF of −0.33 W m$^{-2}$. Despite the small net aerosol DRF (–0.05 W m$^{-2}$) at the TOA, aerosol-radiation interactions could explain the surface brightening over China in the recent decade. The overall reductions in aerosol burdens and associated optical effects mainly from BC and sulfate

enhanced the regional annual-mean downward solar radiation flux at the surface by +1.0 W m$^{-2}$ between 2008 and 2016. The enhancement was in general agreement with a long-term observational record of surface energy fluxes in China. We also estimated that aerosol effects on cloud radiative forcings may have played a dominant role in the net aerosol radiative forcings at the TOA over both China and northern Pacific Ocean during the study period. This study will facilitate more informed assessment of climate responses to projected emissions in the future as well as to sudden changes of human activities (e.g. the

COVID-19 lockdown).

## 1. Introduction

Aerosols perturb the global energy balance through the scattering and absorption of sunlight (Charlson et al., 1992) and serving as cloud condensation nuclei (CCN) particles which impact both the cloud albedo and cloud lifetime (Twomey, 1974; Andreae

and Rosenfeld, 2008). Changes in anthropogenic aerosol concentrations from preindustrial times to present day are estimated to induce a global-mean net cooling effect of $-0.4$ to $-1.5$ W m$^{-2}$ at the top of the atmosphere (TOA) that partly masks the warming effects by increased carbon dioxide concentrations (Boucher et al., 2013).

It is commonly known that black carbon (BC) and sulfate aerosols are important contributors to the radiation absorption and scattering effects of anthropogenic aerosols on a global scale. Bond et al. (2013) have estimated that the industrial-era (1750–2005) direct radiative forcing (DRF) of BC was 0.71 W m$^{-2}$. Until now, the uncertainties embedded in the radiative forcing (RF) of BC are still large because the treatment of BC atmospheric processes in climate models including the impacts of BC on liquid clouds (Koch and Del Genio, 2010; Chung and Seinfeld, 2002) and the role of BC in acting as ice nuclei (Kulkarni et al., 2016) is imprecise. The mixing state of BC is one of the key parameters that determine its optical properties and CCN activity (Jacobson, 2001; Stier et al., 2006; Matsui, 2016). Recent studies have revealed that explicit representation of BC aging processes can reduce the uncertainty on the estimates of BC DRF (Matsui et al., 2018a). Unlike BC, which is emitted directly into the atmosphere, sulfate aerosols originate mainly from the chemical transformation of sulfur dioxide (SO$_2$) via gas-phase photochemical oxidation by the OH radical and by aqueous and heterogeneous reactions (Seinfeld and Pandis, 2016). Estimates of sulfate radiative forcings rely heavily on the representation of secondary sulfate formation in climate models and SO$_2$ emissions (Huang et al., 2015). Sulfate aerosols are estimated to induce a global mean DRF of $-0.32$ W m$^{-2}$ during the period of 1750–2010 (Myhre et al., 2013) and cause a remarkable radiative perturbation in the north mid-latitude region (20 °–40 °N) because of the rapid increase of anthropogenic SO$_2$ emissions in China over the past few decades. The tremendous anthropogenic emissions in China have resulted in not only severe air pollution, but also in a significant alteration of the global shortwave radiation budget (Li et al., 2016a).

China has implemented stringent air pollution control measures during the recent decade and the Chinese SO$_2$ emissions started to decrease in 2007 with the application of flue gas desulfurization in the power plant sector. Especially since 2013, the toughest-ever clean air policies have led to substantial reductions in anthropogenic emissions in China. According to the latest emission inventory, the Chinese annual emissions of SO$_2$, nitrogen oxides (NO$_x$), BC, and organic carbon (OC) declined by 62%, 17%, 27%, and 35%, respectively, during 2010–2017 (Zheng et al., 2018). Recent studies have demonstrated significant improvements of air quality in China attributable to those various emission control measures (Zhang et al., 2019). The reduction of SO$_2$ emissions was particularly noteworthy during this period and it led to a dramatic reduction of sulfate aerosol concentrations and mitigated the problems associated with PM$_{2.5}$ pollution and acid rain (Liu et al., 2020; Liu et al., 2018).

These unprecedented reductions of aerosol and reactive gas emissions over China provide a unique opportunity to investigate climate impacts of such considerable variations in anthropogenic emissions. Those changes of emissions could lead to perturbations of regional solar radiation budgets that are associated with the radiative forcings due to aerosol-radiation interactions and aerosol-cloud interactions. Fadnavis et al. (2019) estimate that the reduction of Chinese SO$_2$ emissions during 2006–2017 produced a positive clear-sky direct radiative forcing of $+0.6$ to $+6$ W m$^{-2}$ over China. Paulot et al. (2018)

investigated the trends of aerosol radiative effects in eastern China from 2001 to 2015 and showed a clear decrease of aerosol optical depths (AODs) starting from 2007 due to reductions of $SO_2$ emissions. We have found that previous estimates of aerosol forcings may have been inadequate because those simulations assumed clear-sky conditions and used a simple treatment of the mixing between sulfate and BC, and did not consider aerosol-cloud interactions, all of which are important in the calculation of aerosol total radiative effects (Ghan, 2013). Until now, it has been unclear how the overall changes of anthropogenic emissions in China (since 2007) including emissions of not only $SO_2$ but also BC and other aerosol components, have impacted aerosol forcings in source regions and outflows. BC emissions are reported to have significantly declined in eastern China from 2010 to 2019 based on the rapid decrease of observed BC concentrations at an in-situ site near mainland China (Kanaya et al., 2020). Because BC particles always exert a positive radiative effect through direct absorption of solar radiation and absorption enhancement by non-BC particles (like sulfate and organics) in the atmosphere (Matsui, 2020), the RFs caused by the changes in the emissions of BC and non-BC particles should be quantified. A comprehensive assessment of aerosol-driven RFs due to the substantial changes in those emissions can also enable better prediction of the regional- to global-scale climate responses to future emission scenarios like the shared-economic pathways (SSPs).

Our goal in this study was to evaluate the response of RFs, including aerosol DRFs and aerosol effects on clouds, to the changes in anthropogenic emissions in China during 2008–2016. We chose this period for analysis because the difference of emissions between 2008 and 2016 can reflect the inter-annual changes in China's anthropogenic emissions caused by a series of national air pollution control measures conducted over the past decade (Li et al., 2017; Zhang et al., 2019). We performed several model experiments with an advanced global climate model integrated with the latest bottom-up emission inventory for China to quantify the RFs from the changes in different aerosol components (e.g., sulfate, nitrate, BC) and their interactions with radiation and clouds. We also predicted the aerosol RFs with one projected emission scenario for the years 2030 and 2050.

## 2 Methods

### 2.1 Model experiments

In this study, we used the Community Atmosphere Model version 5 (Liu et al., 2012; Lamarque et al., 2012) with the Aerosol Two-dimensional bin module for foRmation and Aging Simulation version 2 (CAM5/ATRAS2) (Matsui and Mahowald, 2017; Matsui, 2017; Matsui et al., 2014). The CAM5/ATRAS2 model employed a two-dimensional sectional representation of aerosols with 12 particle size bins (from 1 to 10,000 nm in diameter) and 8 BC mixing state bins (from fresh BC to aged BC-containing particles) for various microphysical and chemical processes of aerosols, including new particle formation, condensation/coagulation, aerosol activation, wet/dry deposition, and interactions with radiation and clouds. The model treats aerosol–cloud interactions in stratiform clouds using a physically based two-moment parameterization that considers aerosol effects on cloud properties (Morrison and Gettelman, 2008). The secondary formation of sulfate has been suggested to be important to the simulation of sulfate concentrations in China, but is not represented well in current chemistry-climate models

(Hung and Hoffmann, 2015; Cheng et al., 2016). To better reproduce the temporal evolution of sulfate concentrations in China, we added a new pathway suggested by previous studies for secondary sulfate formation into our model: the heterogeneous oxidation of gaseous $SO_2$ to particulate sulfate on aerosol surfaces under high-humidity conditions (Huang et al., 2014). The uptake coefficients of $SO_2$ were set to lie in the range $5.0 \times 10^{-5}$ and $5.0 \times 10^{-4}$ under 50–100% ambient relative humidity.

The model was run at a horizontal resolution of $1.9\,° \times 2.5\,°$ with 30 vertical layers from the surface to ~40 km on a global scale. Several simulation experiments were designed with different anthropogenic emission inputs (detailed in section 2.3) as shown in Table 1. We varied the anthropogenic emissions of all species for the years 2008 and 2016 in China (simulations Exp08 and Exp16, respectively). Simulations Exp16SO2 and Exp16BC were made by replacing the $SO_2$ and BC emissions in the Exp08 with the corresponding emissions for 2016. Note that the radiative effects of each aerosol component were calculated online with our model, and the main conclusions of the study were based on the results from the Exp08 and Exp16 cases. The Exp16SO2 and Exp16BC cases, which involved separate changes of the emissions of $SO_2$ and BC, respectively, were used for comparison with the Exp16 case to reveal the interactions of BC and $SO_2$ emissions. We performed two-year simulations for each experiment; the first year was for spin-up, and the second for analysis. The meteorological fields were nudged to the Modern-Era Retrospective analysis for Research and Applications Version 2 data and were fixed at year 2008 in these experiments to separate the emission contribution from those of variations in meteorological fields on aerosol radiative effects. Inter-annual changes in meteorological conditions could influence aerosol radiative forcings but were not the focus of this study, and we simply discussed their importance using the simulation Exp16m (the meteorological fields and emissions at year 2016) in Section 3.3.

**2.2 Calculation of aerosol radiative forcings**

We calculated both the direct radiative effects of aerosols (DRE) and cloud radiative effects (CRE) in each simulation. The changes in DRE and CRE due to changes in anthropogenic emissions over China between 2008 and 2016 can be regarded as aerosol DRF and aerosol-induced cloud radiative forcing (CRF). Specifically, the aerosol DRE at the TOA for each specific aerosol component (e.g., sulfate, BC, and others) was calculated online as the differences between the standard radiation flux and the flux calculated by subtracting that component from the radiation module. The aerosol DRF was then equated to the difference of the DREs between the Exp08 and Exp16 cases.

The aerosol effects on CRF were calculated online following the methodology in Ghan (2013) using the clean-sky (neglecting the scattering and absorption of all aerosol species) radiation flux with and without clouds. To take into considerations the large uncertainty in CRF estimates, we estimated another CRF by scaling the online-calculated CRF based on previously reported CRF estimates. For that estimation, we performed a global simulation using the preindustrial condition (emissions for 1750) and obtained a global and annual mean aerosol CRF (shortwave + longwave) of –2.7 W m$^{-2}$ between the emission years 1750 and 2008, which was much more negative than the corresponding best estimate (–0.5 W m$^{-2}$) in the Fifth Assessment Report of the Intergovernmental Panel on Climate Change (IPCC AR5) (Boucher et al., 2013) as well as in many previous

reports (Bellouin et al., 2020; Zelinka et al., 2014). This large difference likely originated mainly from differences between climate models in processes controlling cloud amounts and lifetime and how aerosols affect cloud albedo (Zelinka et al., 2014; Gryspeerdt et al., 2020). We thus inferred the probably higher (more negative) estimate of the CRF between 2008 and 2016 in this study. We derived a globally constant scaling factor from the ratio of the global industrial-era CRF ($-2.7$ W m$^{-2}$) in our simulations to the IPCC AR5 estimate ($-0.5$ W m$^{-2}$) and used that ratio to scale down the online-calculated CRF globally including China and northern Pacific Ocean. Though this scaling factor may vary between regions, the calculations with and without the scaling yielded a reasonable range of CRFs from our model and previous studies.

## 2.3 Anthropogenic emissions for China

Global anthropogenic emissions were taken from Hoesly et al. (2018) based on the Community Emissions Data System (CEDS) for the year 2008 and open biomass burning emissions from van Marle et al. (2017). We replaced the CEDS anthropogenic emissions in China by the Multi-resolution Emission Inventory (MEIC) for 2008 or 2016, which has provided a more realistic representation of China's emission trends from fossil fuels and biofuels (Zheng et al., 2018). As shown in Table 2, the anthropogenic emissions of several major species, including BC, primary OC, and $SO_2$, were reduced between 2008 and 2016 because of China's clean-air policies. More than 90% of those emission reductions occurred in eastern China (marked in Fig. 1), a region characterized by a high population density and intense economic activities, and subject to stringent air pollution controls. For instance, emission reductions in eastern China between 2008 and 2016 were 57% for $SO_2$, 27% for BC, and 30% for OC.

## 2.4 Observation data

To evaluate the ability of the model to simulate aerosol concentrations, we used monthly measurements of inorganic aerosol chemical components (sulfate and nitrate) from the Acid Deposition Monitoring Network in East Asia (EANET). That database has been widely used for the studies of air pollution and acid deposition in East Asia (Itahashi et al., 2018). The EANET stations used in this study included one site in southern China and nine sites in western and central Japan (Table S1). We also collected the observed concentrations of aerosol compounds from different measurement campaigns, and the yearly averages of those concentrations were used to indicate temporal changes during the study period (Table S1). For comparison, we extracted the simulated aerosol concentrations from the model surface-level horizontal grid closest to the observation sites. In addition, we used observations of vertical BC concentrations profiles from the HIAPER Pole-to-Pole Observations (HIPPO) aircraft campaigns conducted over the northern Pacific Ocean (Wofsy, 2011), where BC concentrations were influenced by East Asian continental sources. The HIPPO data included measurements of tropospheric BC mass concentrations from a single-particle soot photometer (SP2) during 2009–2011 (Schwarz et al., 2013). The monthly mean BC concentrations in the simulation for 2008 were compared with the corresponding observations from the same month in five HIPPO campaigns, although the years selected for the simulations and observations differed (Fig. S1).

In addition, we compared the AOD at 550 nm retrieved by the Multi-angle Imaging Spectroradiometer (MISR) Level-3 product (Garay et al., 2020) and the dark target and deep blue combined AOD at 550 nm by the Moderate Resolution Imaging Spectroradiometer (MODIS) aboard NASA's Terra satellite (Sayer et al., 2014) to the corresponding model results. Both satellite products provided annual or monthly mean AODs at a horizontal resolution of $0.5\,° \times 0.5\,°$ that could reflect temporal changes over China during 2008–2016. We evaluated the aerosol extinction profiles in eastern China using the CALIPSO (Cloud-Aerosol Lidar and Infrared Pathfinder Satellite Observations) Lidar Level 3 Tropospheric Aerosol Profile Product Version 4.20 (Kim et al., 2018). We also used measurements of AOD and single scatter albedo (SSA) from Aerosol Robotic Network (AERONET) stations. The available stations that had long-term data records were Beijing (39.98 °N, 116.38 °E) and Xianghe (39.75 °N, 116.96 °E).

## 3. Results

### 3.1 Changes in aerosol burdens

We first compared the simulated sulfate mass concentrations in the Exp08 and Exp16 cases with corresponding observations (Fig. 1a, b). The sulfate concentrations simulated by the CAM5/ATRAS2 model generally agreed with available observations with respect to the magnitude and spatial patterns. The annual mean concentrations were as high as 30 µg m$^{-3}$ in eastern China in 2008 and decreased to less than 10 µg m$^{-3}$ in most parts in 2016 due to the substantial reductions in $SO_2$ emissions (Table S2). Greater reductions of more than 10 µg m$^{-3}$ were apparent in the northern part of eastern China (Fig. 1a, b), where the $SO_2$ reductions are the most noteworthy in the country (Li et al., 2017). The observed $SO_2$ columns given by Li et al. (2017) indicate severe $SO_2$ pollution (up to 2.0 DU, 1 DU = $2.69 \times 10^{16}$ molecules cm$^{-2}$) in northern China before 2010 and significant mitigation around 2016, when the columns were as low as ~0.5 DU. The model also satisfactorily captured the seasonal cycle of $SO_2$ retrieved by the Aura Ozone Monitoring Instrument (Fig. S2). The change of annual mean sulfate concentrations in eastern China averaged −6.1 µg m$^{-3}$ (−52%) from 2008 to 2016. This change was consistent with another model result (−50%) within the region (Liu et al., 2018). The averaged sulfate mass column burden (i.e., the integrated concentrations between the surface and the top of the model layer) over eastern China decreased by 7.0 mg m$^{-2}$, or by 39% of the mass in 2008 (Fig. 2a, c). Moreover, the sulfate burden over northern Pacific (rectangle in Fig. 2) decreased by an average of 0.46 mg m$^{-2}$ (−19%), reflecting the impact of $SO_2$ emission reductions in China on downwind regions dominated by mid-latitude westerly winds.

We then verified the changes in modeled BC concentrations using measurements from three typical sites located in eastern China and western Japan (Fig. 1c, d). The observed BC concentrations in Beijing and Shanghai have been decreasing over the recent decade (Xia et al., 2020; Wei et al., 2020). During 2008–2016, the annual mean surface BC concentrations decreased from 8.5 to 3.5 µg m$^{-3}$ in Beijing and from ~4.0 to 2.2 µg m$^{-3}$ in Shanghai. Similar reductions were apparent in our simulations; the averaged decrease was 25% over eastern China. The simulations also reproduced the marked decline in surface BC concentrations observed at Fukue Island in western Japan because of the reduction of BC emissions in eastern China (Kanaya

et al., 2020). We also compared the vertical BC mass concentrations profiles in northern Pacific with the BC measurements from the HIPPO campaigns (Fig. S1). The modeled and observed BC concentrations were generally within one order of magnitude of each other over northern Pacific, but there was an overestimation of BC loadings in the upper troposphere that could lead to enhanced BC absorption above clouds. From the Exp08 and Exp16 simulations, the annual mean BC burdens decreased by 0.36 mg m$^{-2}$ (21%) over eastern China and by 0.013 mg m$^{-2}$ (11%) over northern Pacific (Fig. 2b, d). These results suggest that the integration of CAM5/ATRAS2 model and the MEIC emission inventory for China could reproduce the observed decline in sulfate and BC concentrations during the recent decade caused by changes in anthropogenic emissions.

The particulate nitrate simulations also compared well with the measurements (Fig. S3 and Table S2). The regional annual mean nitrate concentrations near the surface were elevated by ~2.0 μg m$^{-3}$ in eastern China in both the Exp16 and Exp16SO2 cases relative to the concentrations in the Exp08 case. Because NH$_3$ emissions were unchanged, the reduction in SO$_2$ emissions was responsible for the increases of particulate nitrate concentrations. The releases of free ammonia into the air as a result of the reduction of ammonium sulfate particles facilitates the thermodynamic partitioning of nitrate toward the aerosol phase (Ansari and Pandis, 1998). Similar increases (1–2 μg m$^{-3}$) of nitrate concentrations within the region have been reported by Liu et al. (2018) for the study period based on a regional chemical transport model. Leung et al. (2020) have also highlighted the close linkage between reduced SO$_2$ emissions and elevated nitrate concentrations in eastern China. In addition, the mean concentrations of ammonium and organic aerosols (OAs) in eastern China were reduced by 0.23 and 0.13 μg m$^{-3}$ in our simulations, respectively. OA concentrations decreased in northern China because of the reduction of primary OA emissions, but they increased in southern China (Fig. S3).

The simulations further revealed that the significant reductions in SO$_2$ emissions would dampen new particle formation and the growth of particles that might serve as CCNs via condensation and coagulation. Figure 3 shows the longitude-height (pressure in hPa) distribution of the changes (%) in the number concentrations of particles with diameters larger than 10 nm (N10) and diagnostic CCN numbers at supersaturation of 0.4% (CCN0.4) along latitude 35 °N, where reductions of sulfate and BC burdens were noteworthy. The N10 concentrations decreased by ~10–30% in eastern China (100–130 °E) from the surface to 300 hPa. The CCN0.4 concentrations decreased both in eastern China (by ~30%) and in the middle troposphere (700–200 hPa) over northern Pacific (by ~10%). According to the Exp16SO2 simulation, these variations in N10 and CCN0.4 were caused primarily by decreases of SO$_2$ emissions and associated sulfate concentrations in eastern China and downwind regions. The decreases of CCN concentrations likely altered cloud properties through their activation into cloud droplets and subsequently changed the radiation budget.

**3.2 Changes in aerosol optical properties**

The pronounced variations of sulfate and BC concentrations between 2008 and 2016 induced changes in aerosol optical properties, i.e., AOD and absorption aerosol optical depth (AAOD). The CAM/ATRAS2 model captured the observed hotspots of AOD at 550 nm (>0.3) over North Africa, eastern China, the tropical Atlantic Ocean, and southwestern Asia based on the

comparison with the MISR and MODIS measurements (Figs. S4 and S5). The annual mean AOD for eastern China in the model (0.27) was close to the MISR AOD (0.34) but lower than the MODIS value (0.41) in 2008. Our comparison of regional AOD magnitude with satellite-observed measurements were similar to the results of previous modeling studies using CAM models (Sockol and Small Griswold, 2017; He et al., 2015). For instance, Sockol and Small Griswold (2017) reported a negative bias of modeled AODs of less than −0.2 over eastern China compared to MODIS observations. Some biases found in continental outflow areas in the tropics and Southern Hemisphere may be related to inadequate simulations of sea salt aerosols and their optical properties in the marine atmosphere (Bian et al., 2019; Burgos et al., 2020).

Figure 4 presents the differences between the Exp08 and Exp16 cases of simulated AOD and AAOD at 550 nm. Corresponding to the reductions in sulfate and BC mass concentrations, AOD decreased by 5–20% over most of eastern China (black box in Fig. 1); the average decrease was 10%. The AOD for sulfate particles decreased by 40% and accounted for much of the AOD variation. Remote sensing observations from the surface (AERONET) and space (MODIS and MISR (Figs. S4 and S5) suggest a similar and statistically significant decrease of the AOD by 10–20% in the same region during the recent decade (Li, 2020; Zhang et al., 2017; Zhao et al., 2017). The regional annual mean AOD retrieved by MISR varied from 0.34±0.07 in 2008 to 0.26±0.06 in 2016 and trended downward during the period (Fig. S6). The AOD simulated by our model decreased on average by about 0.03 between the two years, somewhat less than the MISR observations. This underestimation of AOD changes could be partially attributable to the inadequate representation of decadal variations in dust aerosol concentrations from both natural and anthropogenic sources. A considerable decline in dust emissions has been reported over eastern China during 2007–2014 due to decreased maximum wind speeds and stringent controls of air pollution (Wang et al., 2018), and the resulting decrease of dust loadings and associated light extinction in the atmosphere would have contributed to the decreases of annual mean AOD, but those effects could not be fully represented in our simulations. Moreover, the model parametrizations of aerosol hygroscopicity that determine aerosol extinction coefficients have been an important source of uncertainty on AOD estimates in climate models (Burgos et al., 2020; Reddington et al., 2019).

The simulated annual mean AAOD decreased by about 20% (from 0.018 to 0.014) over eastern China between the emission years 2008 and 2016 (Fig. 4b). A salient decrease as much as 30% occurred in the northern part, where there have been considerable reductions of BC emissions and resultant column burdens (Fig. 2). We compared the model results with the averaged observations at two AERONET stations (located in the same model horizontal grid and marked with a red star in Fig. 4a) with available long-term records. The AAODs at a wavelength of 550 nm at the AERONET site were calculated using the observed AOD and SSA. The percentage decrease of the annual mean AAOD was about 40%, similar to the corresponding model results at the nearest grid (25%). The marked decline in AAOD at this AERONET site was associated with reductions of BC concentrations in urban Beijing (Xia et al., 2020).

We further compared the modeled vertical aerosol extinction profiles with the CALIPSO satellite data over eastern China between 2008 and 2016 during each of the four seasons (Fig. S7). The model could reproduce the shapes of the profiles with

elevated aerosol extinctions ($>0.1$ km$^{-1}$) below altitudes of 2 km above sea level caused by the dense aerosol burdens in the boundary layer. The decrease of aerosol extinctions from 2008 to 2016 was also captured by the simulations. Both the simulation and observations revealed near-surface extinctions as high as 0.4 km$^{-1}$ during the winter, when aerosol pollution was frequent over eastern China, but the observed high extinctions of 0.2–0.4 km$^{-1}$ at near-surface altitudes during summer and fall were underestimated by the model. Because biomass burning activities are predominantly concentrated in the summer and fall in eastern China and are critical determinants of aerosol pollution there (Chen et al., 2017; Wu et al., 2017), the low biases of aerosol extinction profiles could partially stem from the model's use of biomass burning emissions that were derived from the GFED4 database. That emission inventory has been considered to be a lower estimate on a global scale compared to other commonly used databases constrained by satellite observations (Pan et al., 2020). Furthermore, our simulations did not consider inter-annual variations in biomass burning emissions and thus could not reproduce the considerable decline in biomass burning activities and the associated decreases of aerosol extinctions in eastern China over the recent decade (Wu et al., 2020). The severe underestimation of dust extinction profiles against CALIPSO data also contributed to the aforementioned biases (Fig. S8).

Overall, despite the underestimation of AOD changes likely due to inadequate model representation of biomass burning emissions and dust aerosols, our simulations captured a clear decline in aerosol extinctions due to reductions in anthropogenic emissions over China between 2008 and 2016. The resulting effects on the solar radiation budget over China and outflow areas are discussed in section 3.3.

### 3.3 Radiative forcings induced by the changes in anthropogenic emissions between 2008 and 2016

Here, we focus on the all-sky aerosol RFs both at the surface and the TOA that were induced by the inter-annual changes in anthropogenic emissions over China (especially eastern China) between 2008 and 2016. We considered both the aerosol shortwave DRF (also defined as RFs due to aerosol-radiation interactions) and aerosol effects on the shortwave + longwave CRF. As shown in Figs. 5 and S9, the decreases of sulfate mass burdens and their scattering effects due to the substantial reduction of SO$_2$ emissions induced a positive radiative forcing over China. The mean sulfate DRF was +0.58 W m$^{-2}$ over eastern China and regionally as high as +1.0 W m$^{-2}$ in the north part, which experienced the most noteworthy reductions in sulfate burdens. As a result of the reductions of SO$_2$ emissions, nitrate concentrations have increased, and the merged nitrate and ammonium DRF was estimated to be $-0.29$ W m$^{-2}$, though the decrease of particulate ammonium concentrations should yield a positive RF. The net DRF at the TOA from the reduction of SO$_2$ emissions between 2008 and 2016 was therefore +0.29 W m$^{-2}$.

The decline of BC mass burdens diminished their solar absorption over eastern China from 2008 to 2016 (Figs. 5b and S10). The resultant BC DRF there between the Exp08 and Exp16 cases was $-0.33$ W m$^{-2}$ and accounted for 19% of the BC DRE in 2008. Interestingly, this BC DRF was 14% higher than the estimate derived from the simulation scenario with a change of only BC emissions (Exp16BC) because the concurrent declines in sulfate concentrations and associated water uptake weakened

the absorption enhancement of BC-containing particles and thus further diminished the radiative absorption of BC. The absorption enhancements by BC coating materials will be overlooked if BC is externally mixed with other aerosols, which will underestimate the magnitude of the BC DRF for China between 2008 and 2016. Moreover, the treatment of aging processes of BC is another important factor in determining the radiation effects of BC-containing particles. Models with a single mixing state for BC-containing particles (without consideration of the aging processes from fresh to thickly-coated BC) could overestimate (underestimate) the BC DRF when BC-containing particles are assumed to be internally mixed (externally mixed) with other components immediately after emission (Bond et al., 2013; Matsui et al., 2018a).

The net DRF at the TOA due to all aerosol components was −0.05 W m$^{-2}$ over eastern China between the emission years 2008 and 2016. The dominant contributions to this net DRF came from the reductions of $SO_2$ and BC emissions (Fig. 5a). The positive DRF (+0.29 W m$^{-2}$) due to reductions of $SO_2$ emissions was completely offset by the diminished absorption of BC (–0.33 W m$^{-2}$). OA and dust aerosols had the mean DRFs of −0.067 and +3.6 $\times 10^{-4}$ W m$^{-2}$, respectively, and contributed less to the total aerosol DRF. Because of the model underestimation of dust extinction and its temporal changes during the study period, the dust DRF calculated here may have been underestimated. In addition, the total aerosol DRF over northern Pacific was +0.0067 W m$^{-2}$, with main contributions from BC (–0.033 W m$^{-2}$) and sulfate (+0.049 W m$^{-2}$).

The aerosol-induced CRF averaged +0.90 W m$^{-2}$ over eastern China primarily attributable to the decrease of CCN numbers (Fig. 3), implying less solar radiation reflected back to the space by clouds. As mentioned in Section 3.1, the reduction of BC emissions made a negligible contribution to changes of CCN numbers, while the reduction of $SO_2$ emissions and associated decline in sulfate caused an important in-direct effect by serving as CCNs (Twomey, 1974; Andreae and Rosenfeld, 2008). By separating the contributions made by $SO_2$ and BC emissions to CRF (the Exp16SO2 and Exp16BC cases), we found that the annual mean CRF was +0.93 W m$^{-2}$ due to the decrease of $SO_2$ emissions and −0.18 W m$^{-2}$ due to the decrease of BC emissions. Because the net aerosol DRF was relatively small (–0.05 W m$^{-2}$), the aerosol-induced CRF dominated the total RFs at the TOA over eastern China. The northern Pacific had a concurrent regional mean CRF of +0.88 W m$^{-2}$, which was also associated with reductions of $SO_2$ emissions in China.

The adjusted CRFs at the TOA for eastern China and northern Pacific Ocean using a scaling factor of 5.4 based on the IPCC AR5 (see Methods) were +0.17 and +0.16 W m$^{-2}$ and still higher than the aerosol DRF. This scaling factor may vary if we chose other models as reference and the global mean CRF from the IPCC AR5 has been suggested to be a potentially lower (less negative) estimate (Bellouin et al., 2020). The CRF values calculated with and without the scaling here can thus be considered as the lower and upper bounds of the estimates, respectively. In the following analysis, we show only the results with the scaling; those without the scaling can be obtained by multiplying the scaling factor.

This study highlighted the fact that the interactions of anthropogenic aerosols with solar radiation play a key role in enhancing the downward shortwave radiation flux to Earth's surface (surface brightening) in eastern China (Fig. 6). The decreases of BC absorption and sulfate scattering allowed more solar radiation to reach the surface. The changes between the Exp08 and Exp16

cases were estimated to be +0.74 and +0.95 W m$^{-2}$ for BC and sulfate, respectively (Fig. 7b, c). Increases of nitrate concentrations attenuated the downward shortwave flux to the surface. The net change in the annual mean downward radiation flux due to all aerosols was +1.0 W m$^{-2}$ (Fig. 6a). This result is comparable to the long-term observational records of the shortwave energy balance in China from 2000–2015 reported by Schwarz et al. (2020). They suggested a positive trend in downward surface shortwave radiation of +1.4±0.2 W m$^{-2}$ decade$^{-1}$ for that period and attributed it mainly to reductions in the atmospheric shortwave absorption. Our simulations showed that reductions of BC aerosols decreased the atmospheric shortwave absorption by an average of 1.1 W m$^{-2}$ in eastern China for 2008–2016. Apart from BC, other absorbing aerosol components, like natural and anthropogenic dust (Matsui et al., 2018b) and brown carbon from fossil fuels and biomass or biofuel burning sources (Zhang et al., 2020; Yan et al., 2017) may also have contributed to the observed variation of atmospheric shortwave absorption in China and should be considered in future studies. The contribution of aerosol-induced cloud effects on the surface brightening was +0.19 W m$^{-2}$ (with scaling), but this contribution could be masked by increased cloud cover during the study period (Liu et al., 2016). In summary, our simulations provided the first modeling evidence to demonstrate that the mitigation of aerosol pollution, particularly BC and sulfate, has critically determined the surface brightening observed in China over the recent decade (Yang et al., 2019; Schwarz et al., 2020).

To understand how aerosol DRF and CRF respond to future emissions, we referred to the projected anthropogenic emission scenarios in China in 2030 and 2050 recently developed by Tong et al. (2020). For these scenarios, they integrated SSPs, climate targets of Representative Concentration Pathways (RCPs), and local air pollution control measures to create a dynamic projection of China's future emissions. From their designed scenarios, we choose the SSP1-RCP26-BHE (Best Health Effect), which denotes the best-available environmental policies and therefore the largest reductions of emissions, to predict an upper bound on the radiative effects of anthropogenic aerosols. We performed the simulations by using the same meteorological nudging for 2008 (Exp08) and scaled anthropogenic emissions in China separately from 2016 to 2030 and from 2016 to 2050 in accord with Tong et al. (2020) (Table S3). We did not consider the projected emissions of greenhouse gases, and only the aerosol effects on the radiation budget are shown here. We found that under the strictest air pollution control policies, the aerosol DRF estimates were +0.25 W m$^{-2}$ between 2016 and 2030 and +0.66 W m$^{-2}$ between 2016 and 2050 over eastern China (Fig. 7). The total RFs with the combination of DRF and CRF reached +0.52 W m$^{-2}$ (2016–2030) and +1.24 W m$^{-2}$ (2016–2050), respectively. For this scenario, although future BC emissions continued to decrease, the reductions of scattering aerosols like OA and nitrate induced high, positive RFs. The predicted DRFs of OA and nitrate due to aerosol-radiation interactions were +0.28 and +0.42 W m$^{-2}$, respectively, between 2016 and 2050. Samset et al. (2019) estimated a net radiative forcing induced by sulfate and BC aerosols of approximately +0.5 W m$^{-2}$ over China under a strong air-quality policy (SSP1-1.9) during 2014–2030 using the climate model Oslo CTM3 with an oversimplification of aerosol-cloud interaction effects.

This study focuses mainly on the impacts of anthropogenic emission variations on aerosol radiative forcings, while the meteorological conditions could also affect aerosol forcings. We performed another simulation experiment with the meteorological nudging and anthropogenic emissions for 2016 (Exp16m) and compared the resulting aerosol DRF with those

found in the Exp16 case (only emission variation; see Table S4). Note that in the Exp16m case, the model could not separate aerosol effects on clouds from the variation in meteorological fields. We found that both the changes of BC and sulfate mass burdens between the Exp08 and Exp16m cases were close to those between the Exp08 and Exp16 cases, but the BC DRF was reduced by 27% because of the variation in meteorology between 2008 and 2016. The underestimation of AOD temporal changes did not be improved in the Exp16m case. Nevertheless, both our results and previous studies (Liu et al., 2020; Zhang et al., 2019) demonstrated that the decadal variations in aerosol pollution and DRF were dominated by reductions in anthropogenic emissions over the recent decade.

## 4. Summary

Aerosols arising from both anthropogenic and natural sources impact global and regional climate systems through their interactions with radiation and clouds. The unprecedented changes in anthropogenic emissions of reactive gases and aerosols over China in the recent decade were expected to have perturbed the regional radiation budget, yet this impact has not been assessed until now. Here, we quantified the RFs for China (especially eastern China) and its outflow regions induced by changes in anthropogenic emissions in the country between 2008 and 2016 (changes in meteorological conditions were not considered) via a global climate model (CAM5/ATRAS2) and a localized emission inventory (MEIC). Our simulations reproduced well the decreased sulfate and BC mass burdens (by 39% and 21%, respectively) due to the substantial reductions of $SO_2$ and BC emissions (−57% and −27%, respectively). Both the modeled AOD and AAOD showed a clear decline in response to reduced scattering and absorption by aerosols. Our major findings are summarized as follows.

The annual mean instantaneous RF due to aerosol-radiation interaction (difference between the Exp08 and Exp16 cases) was −0.05 W m$^{-2}$ over eastern China. The reduction of $SO_2$ emissions produced a sulfate DRF of +0.58 W m$^{-2}$ and a combined nitrate and ammonium DRF of −0.29 W m$^{-2}$. Previous reports have often focused on the positive DRF by the reduction of $SO_2$ emissions over the past decade, whereas this study found that those effects were completely offset by the simultaneous decrease of BC absorption (−0.33 W m$^{-2}$). Notably, the declines in sulfate concentrations and associated water uptake reduced the absorption enhancement of BC-containing particles and the resulting BC DRF was 14% higher than the DRF with considering only changes in BC emissions. Such an effect could not be detected without an explicit treatment of absorption enhancement for internally mixed BC particles as developed in our model (Matsui, 2020).

While the net aerosol DRF at the TOA was relatively small, aerosol-radiation interactions were responsible for the surface brightening over China in the recent decade. The overall reductions in aerosol burdens (including mainly BC and sulfate) and associated optical effects enhanced the annul mean downward solar radiation flux at the surface by +1.0 W m$^{-2}$ during the study period. This enhancement was comparable to the long-term (2005–2015) observational trend of the downward shortwave radiation flux reaching the Earth's surface (Schwarz et al., 2020). Our results provided the first modeling evidence for the

cause of observed surface brightening in China over the recent decade. Even though the BC and $SO_2$ emissions exerted an almost balanced direct radiative forcing at the TOA, the solar radiation budgets have been perturbed both at the surface and in the atmosphere, which would influence the surface temperature, boundary layer development, and the East Asian monsoon (Huang et al., 2018; Hong et al., 2020; Li et al., 2016b).

Moreover, we estimated that aerosol-induced CRF may dominate the total RFs at the TOA both over China and northern Pacific Ocean. The considerable decreases of sulfate aerosols caused a positive CRF through their influences on liquid cloud droplet numbers and cloud albedo. The online-calculated mean CRFs reached +0.90 and +0.88 W m$^{-2}$ over eastern China and northern Pacific Ocean, respectively. They were scaled down to +0.17 and +0.16 W m$^{-2}$ by a globally constant factor of 5.4. This scaling factor was derived from the ratio of the global annual mean CRF (between present-day and preindustrial time) in our simulations to the corresponding best estimate in the IPCC AR5. Overall, the absolute magnitudes of both of these two estimates (with and without scaling) were appreciably higher than the net aerosol DRF.

China's ambitious targets to further improve air quality and more strongly regulate greenhouse gas emissions will encourage a greater abatement of emissions in the coming decades. We predict that the aerosol RFs at the TOA will reach as high as +0.52 W m$^{-2}$ during 2016−2030 and +1.24 W m$^{-2}$ during 2016−2050 over eastern China under a stringent environmental and climate-mitigation pathway. Our assessment of the historical aerosol radiative forcings in China over the past decade along with the predictions under the future emission scenario will be helpful to better evaluate the climate responses to human activities on regional and global scales.

## Appendix A: List of acronyms

| | |
|---|---|
| AERONET | Aerosol robotic network |
| AOD | Aerosol optical depth |
| AAOD | Absorption aerosol optical depth |
| ATRAS2 | Aerosol Two-dimensional bin module for foRmation and Aging Simulation version 2 |
| BC | Black carbon |
| BHE | Best health effect |
| CALIPSO | Cloud-Aerosol Lidar Infrared Pathfinder Satellite Observations |
| CAM5 | Community Atmospheric Model version 5 |
| CCN | Cloud condensation nuclei |
| CCN0.4 | CCN number concentrations at supersaturation of 0.4% |
| CRF | Cloud radiative forcing |
| DRE | Direct radiative effect |
| DRF | Direct radiative forcing |

| | | |
|---|---|---|
| EANET | Acid deposition monitoring network in East Asia | |
| HIPPO | HIAPER Pole-to-Pole Observations | |
| IPCC AR5 | The Fifth assessment report of the Intergovernmental Panel on Climate Change | |
| MEIC | Multi-resolution emission inventory for China | |
| MERRA2 | Modern-Era retrospective analysis for research and applications version 2 | |
| MISR | Multi-angle imaging spectroradiometer | |
| MODIS | Moderate Resolution Imaging Spectroradiometer | |
| N10 | Number concentrations or particles with diameters larger than 10 nm | |
| $NO_x$ | Nitrogen oxides | |
| OC | Organic carbon | |
| OA | Organic aerosol | |
| OMI | Ozone Monitoring Instrument | |
| RCP | Representative Concentration Pathways | |
| RF | Radiative forcing | |
| $SO_2$ | Sulfur dioxide | |
| SSA | Single scatter albedo | |
| SSP | Shared socio-economic pathways | |
| TOA | Top of the atmosphere | |

**Data availability**. EANET observation data is freely available from the online website https://monitoring.eanet.asia/document/public/index. AERONET data for AOD can be accessed from Goddard Space Flight Center (https://aeronet.gsfc.nasa.gov/new_web/data.html). MISR Level-3 data are available from the Atmospheric Science Data Center at NASA (https://asdc.larc.nasa.gov/project/MISR) (Garay et al., 2020). Monthly MODIS AOD data are publicly available from NASA Earth Observations (NEO) (https://neo.sci.gsfc.nasa.gov/view.php?datasetId=MODAL2_M_AER_OD) (Sayer et al., 2014). HIPPO observations of BC vertical profiles are publicly available from the Earth Observing Laboratory (EOL) HIPPO data archive (http://www.eol.ucar.edu/projects/hippo/) (Schwarz et al., 2013). OMI $SO_2$ columns are publicly available from NASA AVDC (https://avdc.gsfc.nasa.gov/pub/). Historical and future anthropogenic emissions for China are publicly available from the MEIC database (http://www.meicmodel.org/) (Zheng et al., 2018). CALIPSO data is publicly available at https://www-calipso.larc.nasa.gov/ (Kim et al., 2018).

**Author contribution.** ML and HM designed the study and wrote the paper. ML performed the simulations and analyzed the results.

**Competing interests.** The authors declare that they have no conflict of interest.

**Acknowledgements.** This work was supported by the Ministry of Education, Culture, Sports, Science, and Technology and the Japan Society for the Promotion of Science (MEXT/JSPS) KAKENHI Grant Numbers JP17H04709, JP19H04253, JP19H05699, JP19KK0265, JP20H00196, and JP20H00638, and by the MEXT Arctic Challenge for Sustainability (ArCS, Program Grant Number JPMXD1300000000) and ArCS-II projects (JPMXD1420318865). This work was also supported by the Environment Research and Technology Development Fund 2–1703 (JPMEERF20172003) and 2-2003 (JPMEERF20202003) of the Environmental Restoration and Conservation Agency. The authors also thank the National Oceanic and Atmospheric Administration (NOAA) Black Carbon Group for providing us their BC data from aircraft measurements.

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

**Table 1**. Model simulation experiments with the differences of anthropogenic emissions for China between the years 2008 and 2016. $SO_2$, sulfate dioxide; BC, black carbon; Others, the anthropogenic emissions of reactive gases and aerosols other than BC and $SO_2$.

| Cases | $SO_2$ | BC | Others | Meteorology |
|---|---|---|---|---|
| Exp08 | 2008 | 2008 | 2008 | 2008 |
| Exp16 | 2016 | 2016 | 2016 | 2008 |
| Exp16SO2 | 2016 | 2008 | 2008 | 2008 |
| Exp16BC | 2008 | 2016 | 2008 | 2008 |
| Exp16m | 2016 | 2016 | 2016 | 2016 |

**Table 2**. Anthropogenic emissions for major species in China and eastern China (EC) between 2008 and 2016 (units: Tg year$^{-1}$). The emissions were derived from the MEIC database (Zheng et al., 2018). The geographical range of the EC is shown in Fig. 1. BC, black carbon; $NO_x$, nitrogen oxides; OC, organic carbon; $PM_{2.5}$, particulate matter with aerodynamic diameter less than 2.5 μm; $SO_2$, sulfur dioxide.

| | 2008 | | 2016 | |
|---|---|---|---|---|
| | China | EC | China | EC |
| BC | 1.7 | 1.5 | 1.3 | 1.1 |
| $NO_x$ | 24 | 21 | 23 | 19 |
| OC | 3.2 | 2.7 | 2.3 | 1.9 |
| $PM_{2.5}$ | 12 | 11 | 8.1 | 6.9 |
| $SO_2$ | 30 | 28 | 13 | 12 |

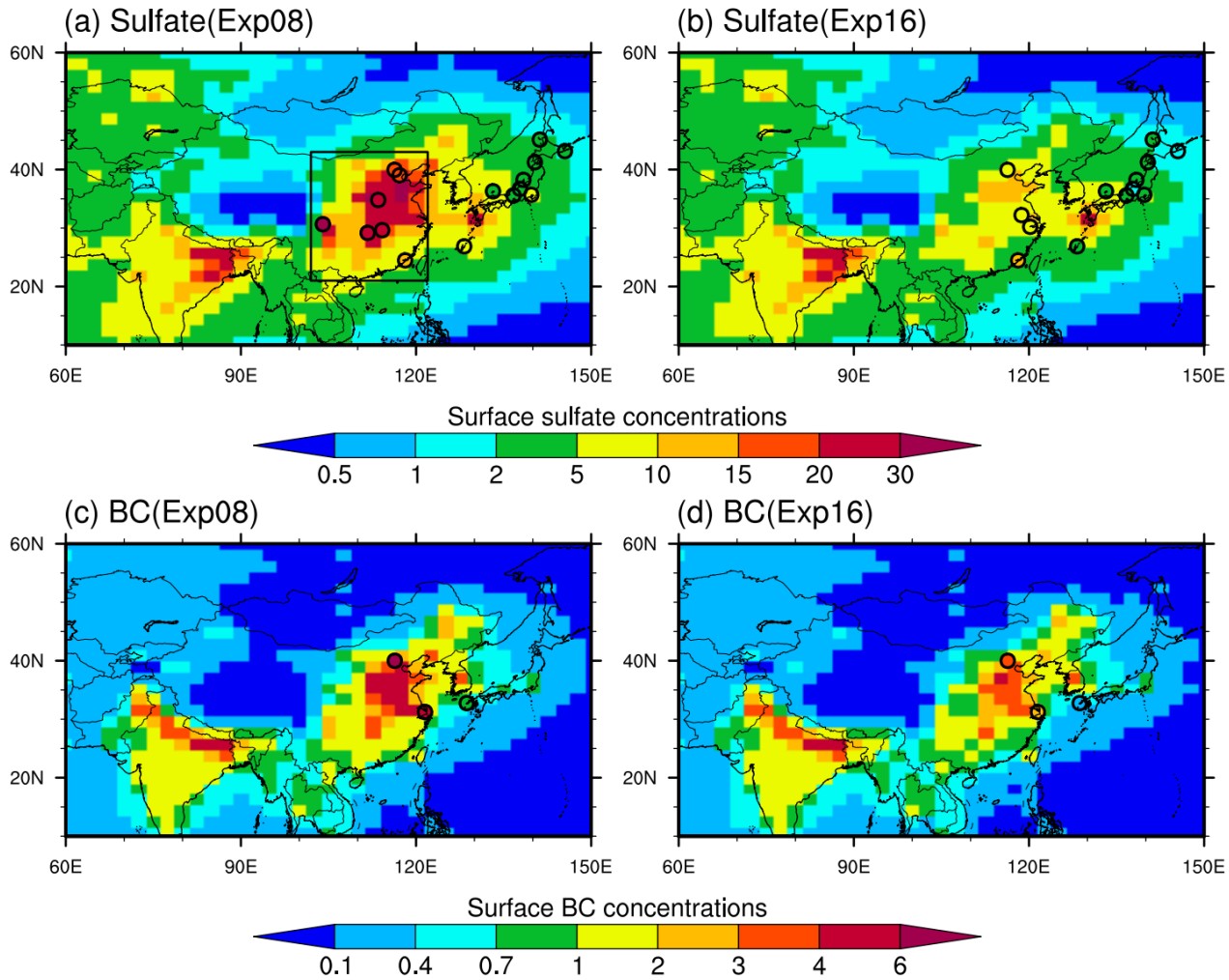

**Figure 1**. Map of simulated surface PM$_{2.5}$ sulfate (a-b) and BC (c-d) concentrations (units: µg m$^{-3}$) over China and surrounding areas in the Exp08 (emissions for the year 2008) and Exp16 (emissions for 2016) cases. Only the changes in anthropogenic emissions over China were considered in these simulations, and the temporal changes of aerosol concentrations may be biased in South Asia where observed aerosol burdens likely increased (Fig. S4). The observations of the annual mean sulfate and BC concentrations obtained from the EANET and literature (Table S1) are overlaid on the map of simulations for comparison. The geographical range of eastern China (21°−43° N, 102°−122° E) is marked in Fig. 1a. The annual mean surface BC concentrations observed at Fukue island (32.75 ºN, 128.68 ºE) in western Japan for 2010 (Fig. 1c) and 2016 (Fig. 1d) are marked. PM$_{2.5}$, particulate matter with aerodynamic diameter less than 2.5 µm; BC, black carbon; EANET, Acid Deposition Monitoring Network in East Asia.

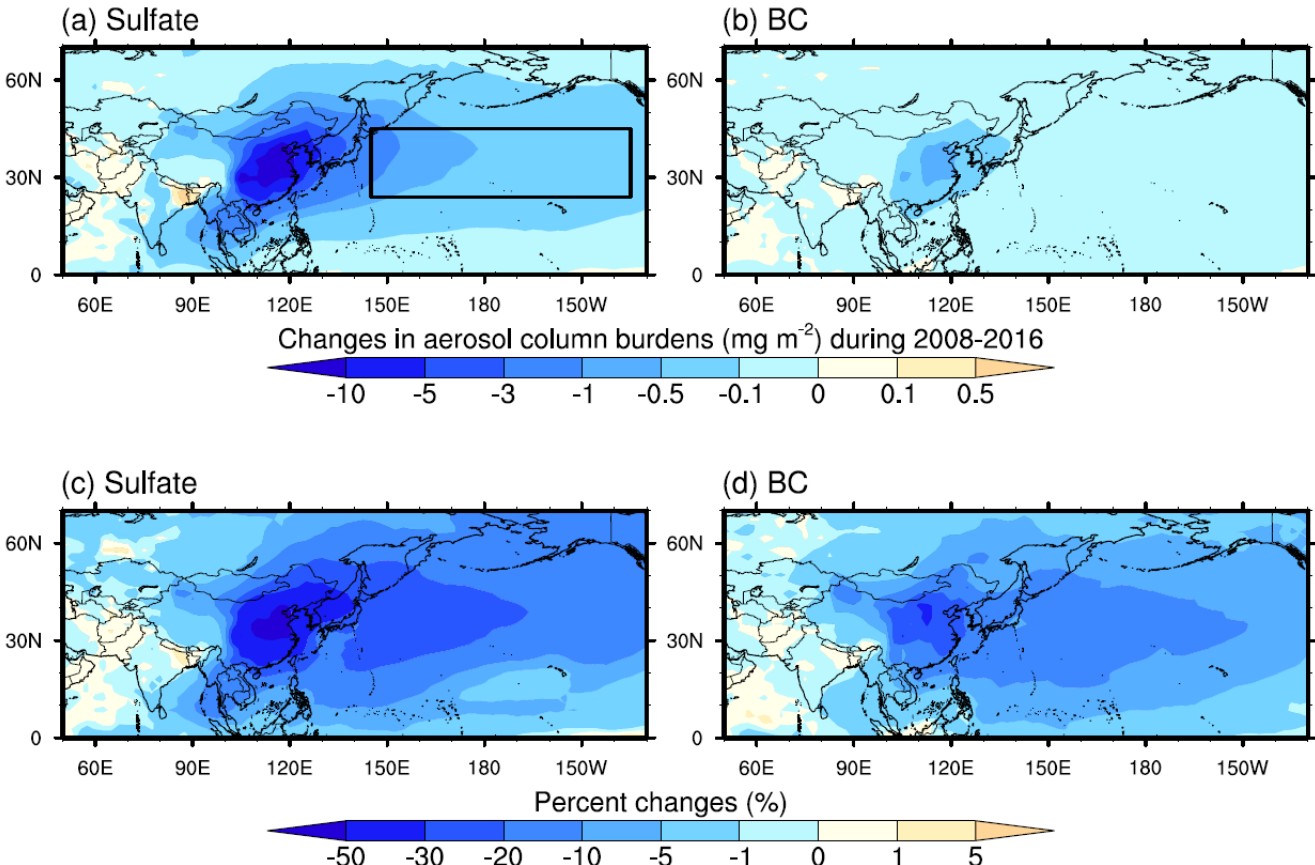

**Figure 2**. Map of the changes in particulate sulfate and BC column burdens due to the anthropogenic emission changes over China between 2008 and 2016. (a-b) and (c-d) denote the mass (mg m$^{-2}$) and percent changes (%), respectively. The northern Pacific Ocean (24°−45° N, 145° E−135 °W) is marked with the black box in Fig. 2a. BC, black carbon.

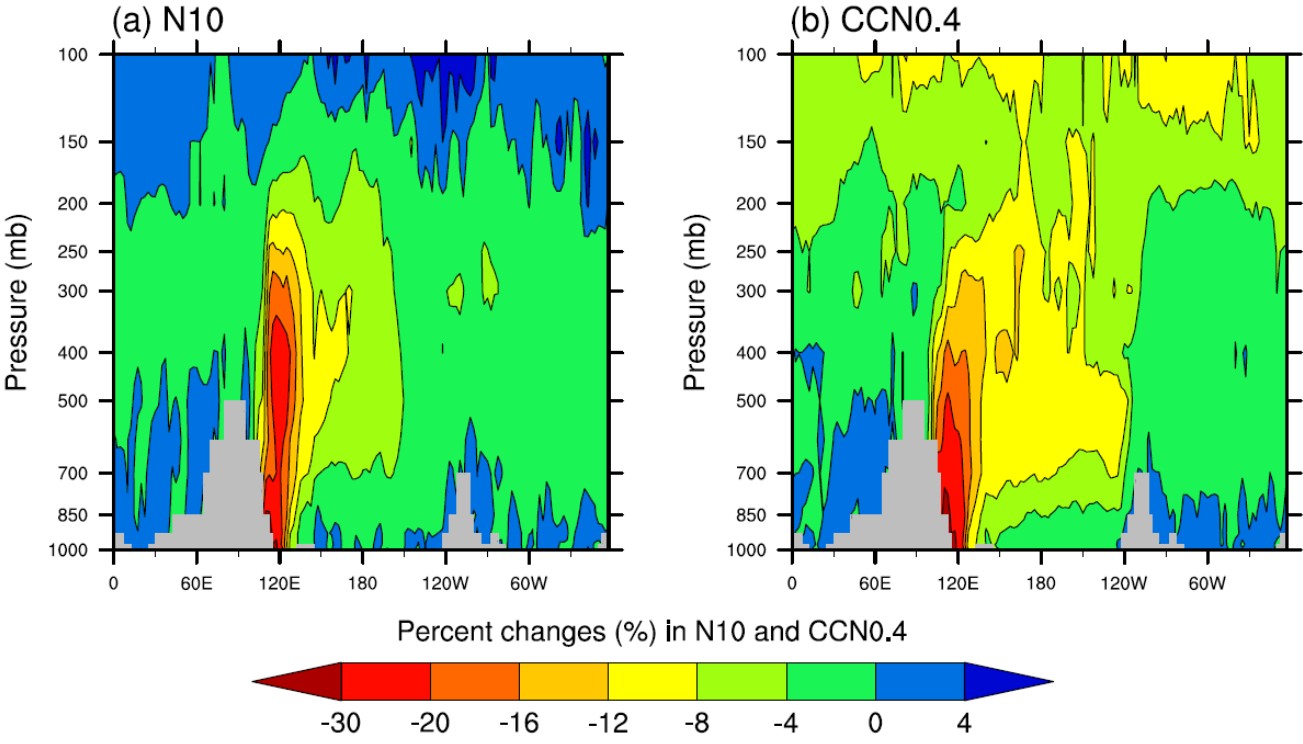

Figure 3. Pressure-longitude distributions of the percent differences of the N10 and CCN0.4 along latitude 35 °N between the Exp08 and Exp16 simulations. N10, particle number concentrations with the diameters larger than 10 nm; CCN0.4, diagnostic CCN number concentrations at supersaturation of 0.4%.

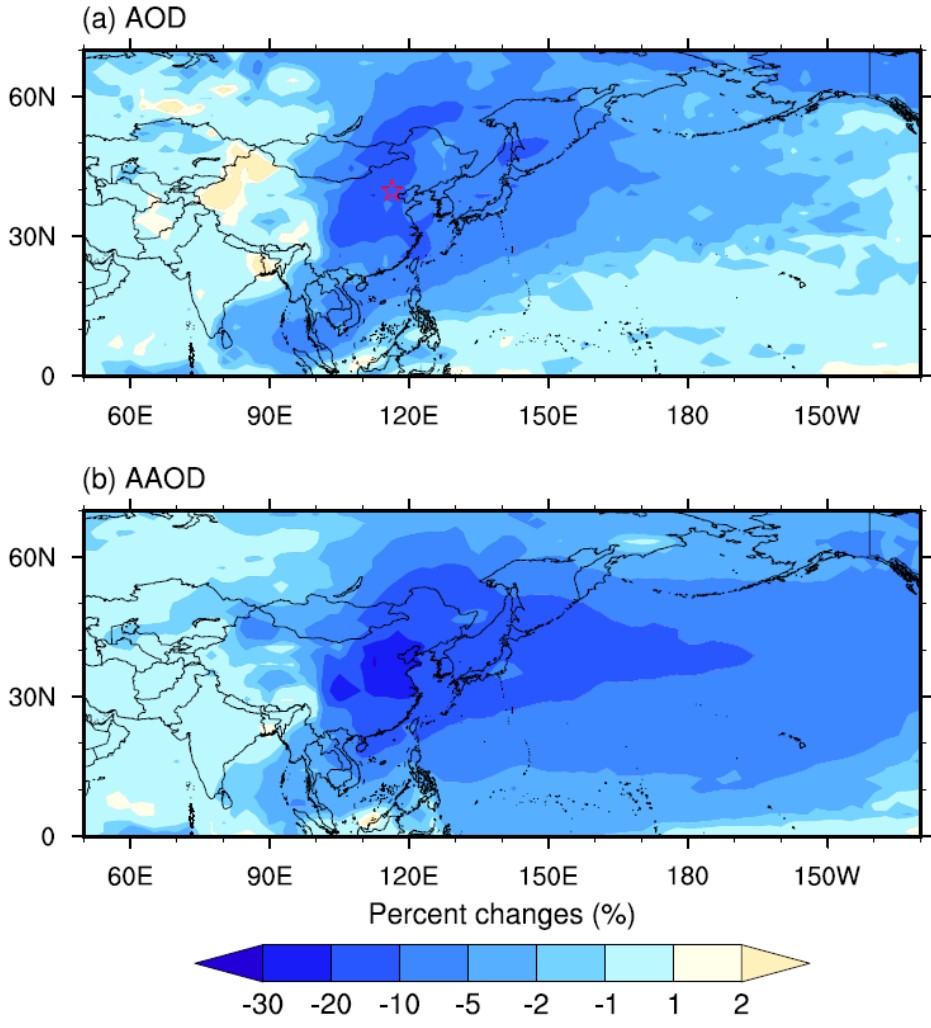

**Figure 4**. Map of the changes (%) in AOD (a) and AAOD (b) due to changes in anthropogenic emissions over China between 2008 (Exp08) and 2016 (Exp16). The location of the AERONET site is marked with a red star. AOD, aerosol optical depth; AAOD, absorption aerosol optical depth.

705

710

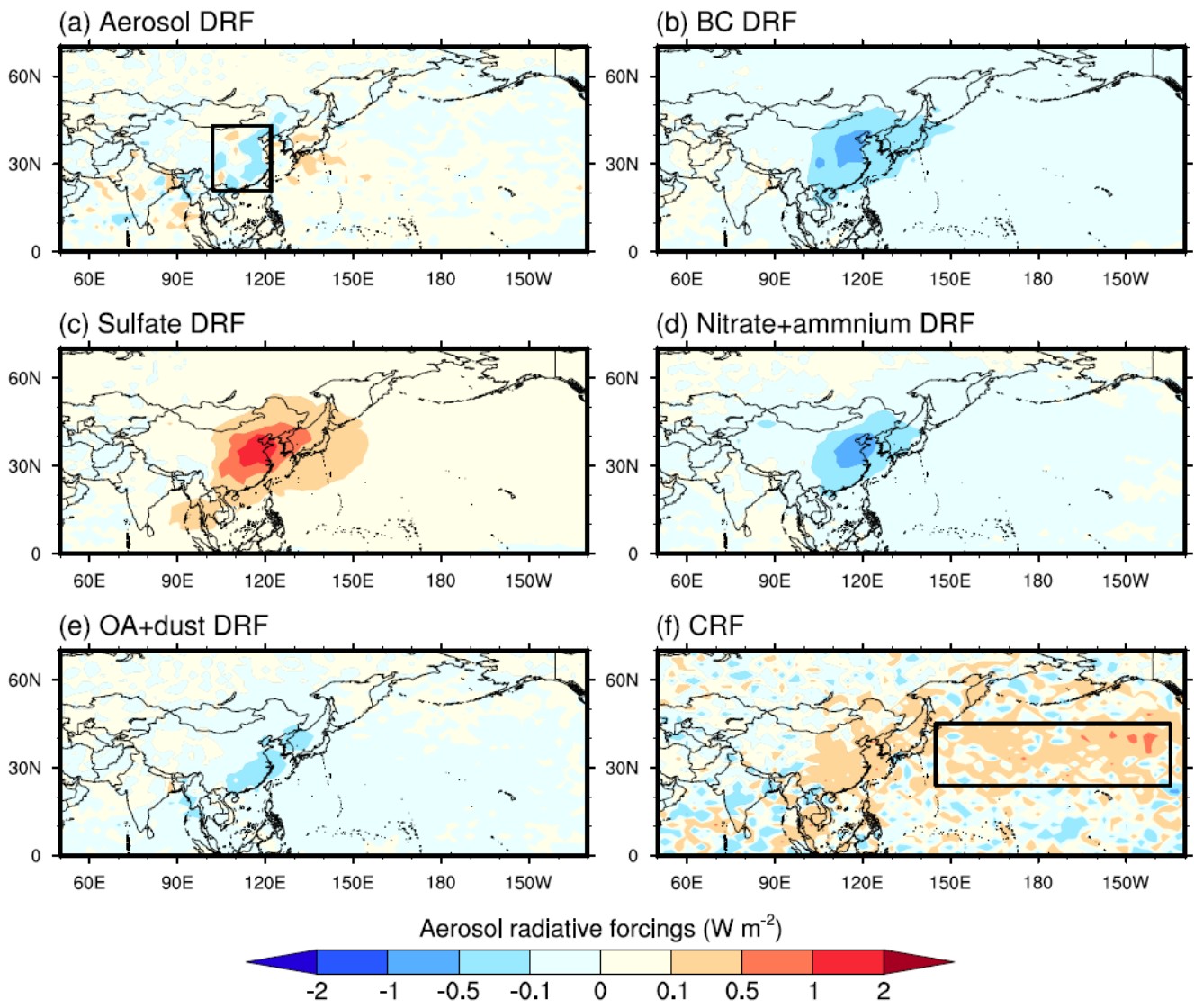

**Figure 5.** Map of DRF from total aerosols (a), BC (b), sulfate (c), nitrate + ammonium (d), and OA + dust (e), and aerosol-induced CRF (f). The CRF values have been scaled by a constant factor of 5.4. Two regions of interest, eastern China and northern Pacific, are marked in Fig. 6a and 6f with black boxes. DRF, direct radiative forcing; CRF, cloud radiative forcing. BC, black carbon; OA, organic aerosols.

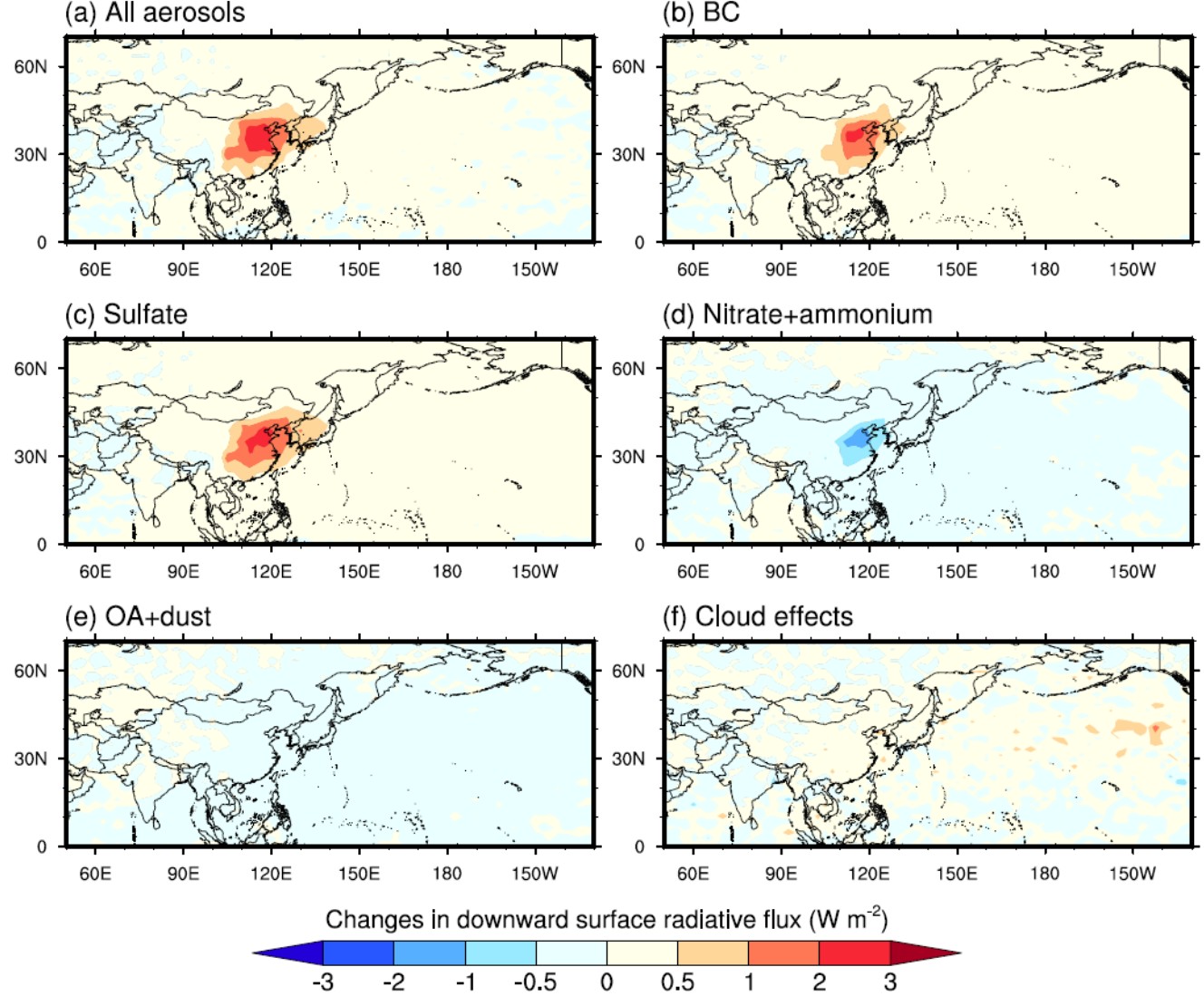

**Figure 6.** Map of the changes in downward shortwave radiation fluxes at the surface over China and outflow regions due to effects of all aerosols (a), BC (b), sulfate (c), nitrate + ammonium (d), OA + dust (e), and clouds induced by aerosols (f). The positive values indicate the surface brightening and the negative surface dimming. BC, black carbon; OA, organic aerosols.

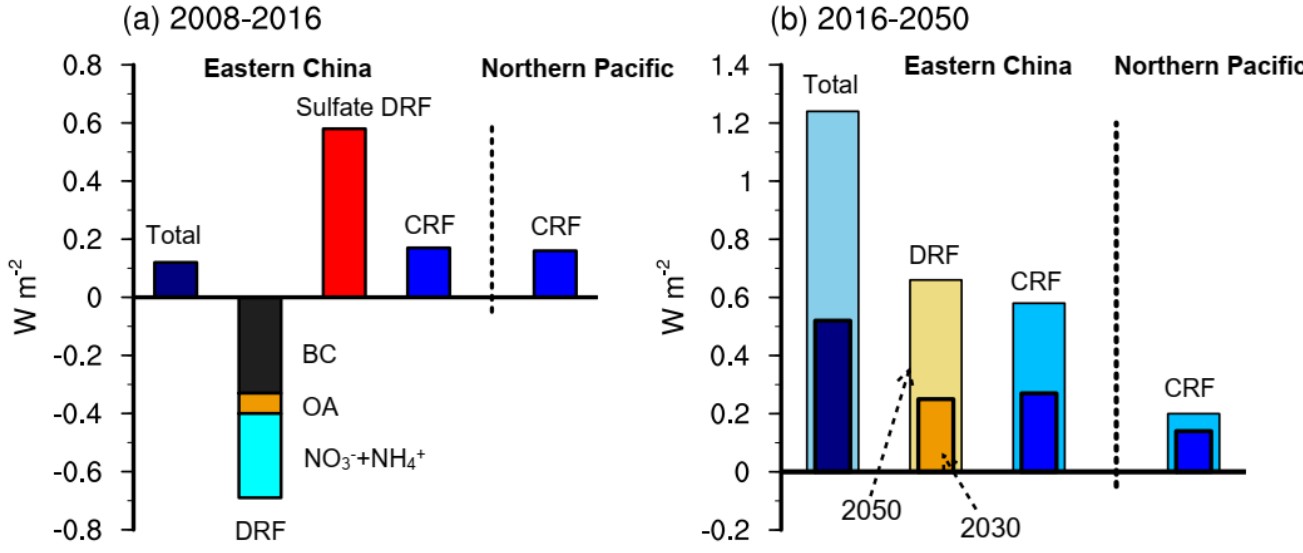

**Figure 7.** (a) The estimates of total RFs (aerosol DRF + CRF), DRF from major aerosol components due to aerosol-radiation interactions, and aerosol effects on clouds (CRF) over eastern China and northern Pacific due to the changes in anthropogenic emissions between 2008 and 2016; (b) The estimates of aerosol DRF and CRF due to projected anthropogenic emissions for 2030 and 2050 relative to 2016. The aerosol DRF over northern Pacific was negligible and not shown here.

RFs, radiative forcings; DRF, direct radiative forcing; CRF, cloud radiative forcings.