# Peer review of "Aerosol radiative forcings induced by the substantial changes in anthropogenic emissions over China during 2008–2016"

_Atmospheric Chemistry and Physics, 2020_

## Referee Comment (RC1) · Anonymous Referee #1 · 3 Sep 2020

One global model is used to study the impact of changes in anthropogenic aerosol emissions in China on regional radiation budget. The authors found that, compared to 2008, reductions in black carbon and sulfur dioxide emissions in 2016, separately, had an opposite effect on radiative forcing, resulting in a net radiative forcing of -0.04 W/m2 by the changes in black carbon and sulfur dioxide emissions. The main scientific conclusion of this study is by no means novel. A number of studies in the literature have shown similar conclusions over China, East Asia and/or other regions around the world. In addition, I have several major specific concerns regarding the methodology

and model biases as described below.

Major comments:

1. "the recent decade" in the title and abstract (and a few other places in the main text) should be changed to the exact time period "2008-2016". Why are these two specific years chosen? Many published studies on aerosol radiative forcing use 2014 as "present day". On the other hand, the simulations were only performed for two years with one year discarded and the results were mostly presented as the difference between the two time slices. I don't think the use "trend" in the description of discussion of results is appropriate. Given the CAM experiment design, comparison of model results in 2016, which is based on meteorological conditions in 2008, to observations is not apples-to-apples. Meteorology is probably quite important for interannual variability in aerosol concentrations (e.g., Yang et al., 2018). Please discuss how this setup could affect your conclusions.

2. I also have a concern about the current experimental design and forcing estimates, which cannot be used to support one of the main conclusion (i.e., the net BC and sulfate radiative forcing of -0.04 W m-2). Forcing induced by BC emissions reduction and forcing induced by SO2 reduction are diagnosed in separate simulations with one of the individual aerosol changes included, assuming that the forcings induced by the individual aerosol changes are additive. Results in the current study and some previous studies (e.g., Chen et al., 2017) have shown nonlinearities between emission and forcing changes even for the same species. On the other hand, the CRF in this study is calculated by scaling results of different simulations with the best estimate of effective radiative forcing in IPCC AR5, which needs to be described more clearly. I am not sure how exactly this was done and why this number is comparable to the DRF calculated in this study. As far as I know, the AR5 effective radiative forcing is calculated differently from the method of Ghan (2013). The latter attributes radiation changes induced by above-cloud light absorbing aerosols to DRF.

3. It looks to me that the CAM AOD is so much off comparing to MISR (Figure S2), especially over the continental outflow regions and the high latitudes. A difference plot would be able to show the biases more clearly. There are various studies in the literature that have shown improved aerosol simulations in CAM5 (e.g., Ghan et al. 2016 and references therein). Why aren't those improvements included in this version of the CAM model? Model biases in AOD as well as the vertical distribution of aerosols and how they can impact the forcing estimates in the two regions of interest need to be quantitatively evaluated and more objectively discussed.

Minor comments and technical edits:

Line 57-59: I believe there are quite a few other studies in the literature that are relevant here.

Line 71: CAM5 should be spelled out as "Community Atmosphere Model version 5"

Line 80: Change "simulation" to "simulating"

Line 108: How about global emissions for 2016? No changes in emissions of other regions? The CEDS emissions only went up to 2014.

Line 127-128: Why MISR AOD is used here rather than other remote sensing or hybrid products that are often used by the aerosol modeling community to evaluate models.

Line 164-165: this sentence needs clarification. Do you mean less ammonium sulfate leads to more ammonium nitrate over East China in 2016? Assuming that all else being equal?

Figure 4: It's very difficult to see the AERONET observations in the figure. Is there only one site that can represent East Asia? I don't see much value added here.

Figure 6: How was the total RF calculated? How does this compared to the TOA forcing difference between Exp08 and Exp16?

References:

Chen, Y., Wang, H., Singh, B., Ma, P. L., Rasch, P. J., & Bond, T. C. (2018). Investigating the linear dependence of direct and indirect radiative forcing on emission of carbonaceous aerosols in a global climate model. Journal of Geophysical Research: Atmospheres, 123, 1657– 1672.

Ghan, S. J., Liu, X., Easter, R. C., Zaveri, R., Rasch, P. J., Yoon, J. H., & Eaton, B. (2012). Toward a minimal representation of aerosols in climate models: Comparative decomposition of aerosol direct, semidirect, and indirect radiative forcing. Journal of Climate, 25(19), 6461–6476.

Ghan, S. J., Wang, M., Zhang, S., Ferrachat, S., Gettelman, A., Griesfeller, J., et al. (2016). Challenges in constraining anthropogenic aerosol effects on cloud radiative forcing using present‐day spatiotemporal variability. Proceedings of the National Academy of Sciences of the United States of America, 113(21), 5804–5811. https://doi.org/10.1073/pnas.1514036113

Yang, Y., Wang, H., Smith, S. J., Zhang, R., Lou, S., Qian, Y., Ma, P.‐L., & Rasch, P. J. (2018). Recent intensification of winter haze in China linked to foreign emissions and meteorology. Scientific Reports, 8, 2107. https://doi.org/10.1038/s41598‐018‐20437‐7

---

## Referee Comment (RC2) · Anonymous Referee #2 · 22 Sep 2020

The manuscript is very well written, both in terms of language usage and in terms of clarity of wordings, it makes a very easy read. However, there are major issues with the presentation of the results, their mis-presentation in some cases -see annotated pdf. I strongly recommend to the authors to consider these comments and improve the presentation of their results in a more scientific manner.

<hr>

---

## Author Comment (AC1) · 27 Oct 2020

**Response to Referee #1**

We sincerely appreciate the Referee #1's comments, which indeed help use to improve the manuscript. We accept all the suggestions proposed by the Referee and revise the manuscript accordingly. Please see the point-point responses to the comments in the following.

*Referee: "One global model is used to study the impact of changes in anthropogenic aerosol emissions in China on regional radiation budget. The authors found that, compared to 2008, reductions in black carbon and sulfur dioxide emissions in 2016, separately, had an opposite effect on radiative forcing, resulting in a net radiative forcing of -0.04 W/m2 by the changes in black carbon and sulfur dioxide emissions. The main scientific conclusion of this study is by no means novel. A number of studies in the literature have shown similar conclusions over China, East Asia and/or other regions around the world. In addition, I have several major specific concerns regarding the methodology and model biases as described below"*

**Response:** Accepted. We improve the description of the novelty and the scientific implications of this study in the new text. The novelty mainly includes:

1) The clean-air measures in China led to the heaviest-ever reductions in anthropogenic emissions over the past decade (please see Table 1 in our manuscript), which provides a unique opportunity to investigate the aerosol radiative forcings due to such substantial variations in emissions. We quantified the aerosol-induced radiative forcings at TOA due to both aerosol-radiation interactions and aerosol-cloud interactions over China and the outflow regions (North Pacific) using a global climate model and the latest China's anthropogenic emission inventory (replacing CMIP6 emissions for China). To our knowledge, no modeling studies have shown such results before. Please see Line 60-77 in the revised manuscript (Unless noted otherwise, line numbers are shown for the revised manuscript).

Our results will implicate for predicting climate impacts of emission variations under different SSP (shared socio-economic pathways) scenarios in the future (e.g., Samset et al., 2019). Please see Section 3.3 (Line 320-337).

2) Recent studies mainly emphasized the radiative forcings of $SO_2$ emission variations over China in the past decade (Fadnavis et al., 2019), but didn't uncover their

interactions with BC and other aerosol species (e.g., nitrate) in determining the net aerosol radiative forcings. Our results suggest that the reductions in $SO_2$ emissions and the associated sulfate concentrations would facilitate the formation of ammonium nitrate and also diminish the radiative forcing of BC-containing particles through the absorption enhancement effect. These complex effects should be considered in the assessment of aerosol radiative forcings. Please see Line 257-264.

3) This study indicates that the reductions in sulfate and BC led to the increase in downward solar radiation flux onto the Earth's surface. The results provide the first modeling evidence for the cause of surface brightening in China over the past decade, as shown by the long-term observations (Schwarz et al., 2020).  Please see Line 301-319.

*Major comments*

*Referee: "1. "the recent decade" in the title and abstract (and a few other places in the main text) should be changed to the exact time period "2008-2016". Why are these two specific years chosen? Many published studies on aerosol radiative forcing use 2014 as "present day". On the other hand, the simulations were only performed for two years with one year discarded and the results were mostly presented as the difference between the two time slices. I don't think the use "trend" in the description of discussion of results is appropriate."*

**Response:** Accepted. We use "2008–2016" in the revised title.

We chose these two years for analysis because the differences of emissions between these two years (2008 and 2016) can reflect the decadal changes in China's anthropogenic emissions caused by a series of emission control policies from 2008 to the present. The $SO_2$ emissions in China peaked around 2007 and have declined since then (Li et al., 2017). And the toughest-ever clean-air policies were conducted between 2013–2017 and led to further reductions in primary aerosol emissions (Zhang et al., 2019). We added more description on the study period in the revised manuscript. Please see Line 79-83.

We revise the statement with the use "trend" in the main text. Some trend results from previous studies and long-term observation data have been shown to support our simulation results. Please see Line 162-183 and Fig. S6.

*Referee: "Given the CAM experiment design, comparison of model results in 2016, which is based on meteorological conditions in 2008, to observations is not apples-to-apples. Meteorology is probably quite important for interannual variability in aerosol concentrations (e.g., Yang et al., 2018). Please discuss how this setup could affect your conclusions."*

**Response:** Accepted. We performed another simulation using both the emissions and meteorology fields for 2016 and discussed how meteorology conditions affect aerosol concentrations and the associated radiative effects. Please see Line 338-346 and Table S4 in the revised manuscript and SI.

Both this study and previous studies demonstrate that the substantial reductions in anthropogenic emissions dominated the mitigation of aerosol pollutions in China in the past decade (Liu et al., 2020; Zhang Q. et al., 2019; Zhang X., et al., 2019). Variation in meteorological conditions only has a minor effect on the decadal variability in aerosol concentrations and their radiative effects due to aerosol-radiation interactions over China (Table S2).

Note that the aerosol-induced cloud radiative forcing cannot be separated from the changes of meteorological fields. This study mainly focuses on the impacts of inter-annual variations in anthropogenic emissions on aerosol radiative effects and meteorological factors are not the key topic of this study.

*Referee: "2. I also have a concern about the current experimental design and forcing estimates, which cannot be used to support one of the main conclusion (i.e., the net BC and sulfate radiative forcing of -0.04 W m-2). Forcing induced by BC emissions reduction and forcing induced by SO2 reduction are diagnosed in separate simulations with one of the individual aerosol changes included, assuming that the forcings induced by the individual aerosol changes are additive. Results in the current study and some previous studies (e.g., Chen et al., 2017) have shown nonlinearities between emission and forcing changes even for the same species."*

**Response:** In this study, the forcings induced by BC and SO$_2$ emissions were calculated online in the same simulations (i.e., differences between Exp16 and Exp08). The separate simulations (mentioned by the referee) with one of the emission change included were only used for comparison instead of drawing the main conclusion. Therefore, the nonlinearities between emissions and forcings have been considered in our study. We add more description for the calculation of aerosol forcings. Please see Line 103-115.

*Referee: "On the other hand, the CRF in this study is calculated by scaling results of different simulations with the best estimate of effective radiative forcing in IPCC AR5, which needs to be described more clearly. I am not sure how exactly this was done and why this number is comparable to the DRF calculated in this study. As far as I know, the AR5 effective radiative forcing is calculated differently from the method of Ghan (2013). The latter attributes radiation changes induced by above-cloud light absorbing aerosols to DRF."*

**Response:** Accepted. In this study, we scaled the global-mean CRF (aerosol effects on cloud forcings) calculated in our model to the best estimate of the effective radiative forcing due to aerosol-cloud interactions ($-0.5$ W m$^{-2}$) given in Chapter 7 of IPCC AR5 report (Boucher et al., 2013), which makes our CRF results consistent with the current best estimate of CRF. We do not scale the DRF estimate in this study. The DRF is online diagnosed in the model using the method of Ghan (2013) that attributes the above-cloud light absorbing effects by aerosols to DRF. We reword the description for the calculation of CRF in this study. Please see Line 116-126.

*Referee: "3. It looks to me that the CAM AOD is so much off comparing to MISR (Figure S2), especially over the continental outflow regions and the high latitudes. A difference plot would be able to show the biases more clearly. There are various studies in the literature that have shown improved aerosol simulations in CAM5 (e.g., Ghan et al. 2016 and references therein). Why aren't those improvements included in this version of the CAM model? Model biases in AOD as well as the vertical distribution of aerosols and how they can impact the forcing estimates in the two regions of interest need to be quantitatively evaluated and more objectively discussed."*

**Response:** Accepted. We add more discussions on the model bias in AOD, compare the vertical distribution of modeled BC concentrations with aircraft measurements, and evaluate the uncertainties in the forcing estimates. Here, we point out that:

1) A difference plot between modeled and observed AOD from MISR and MODIS have been given in the Supplementary Materials (Figure S5). The model biases in AOD over some continental outflow regions and high latitudes (mainly in Southeastern Asia and East of Africa) are commonly found in CAM models (Sockol and Griswold, 2018; He et al., 2015), even if improvements for aerosol simulations have been included. It's quite difficult to quantitatively evaluate the model biases of radiative forcing estimates, but a qualitatively description of the potential bias related to AOD has been added in the revised manuscript. Please see Line 225-236 and Line 274-279.

2) The aerosol simulations have been markedly improved by our group based on standard CAM models (Matsui 2017; Matsui and Mahowald, 2017). Our model uses an advanced two-sectional bin representation of size and mixing states of BC particles, which calculates detailed aerosol processes and their interactions with radiation and clouds. We also include a new pathway of secondary sulfate formation as well as an improved volatility basis-set approach for secondary organic aerosols. Our simulations can generally reproduce the spatiotemporal features of major aerosol compositions over China and outflow areas (Figure 2, Figure S3, and Table S2). In our understanding, the paper mentioned by the referee, i.e., Ghan et al. (2016), does not discus improvements of aerosol simulations.

3) Because the observations of vertical profiles of sulfate and BC for a long period (> 1 month) are quite limited in China mainland, we evaluate the model performances of BC profiles in the continental outflow region (i.e., North Pacific) using the aircraft measurements from the HIPPO campaign. Please see Figure S1 and Line 183-186 in the revised manuscript.

*Minor comments and technical edits:*

*Referee: "Line 57-59: I believe there are quite a few other studies in the literature that are relevant here."*

**Response:** Accepted. We add relevant literatures in the revised manuscript. To our knowledge, very few studies have focused on the aerosol radiative forcings in response to the substantial emission variations in China over the past decade. Please see Line 61-72.

*Referee: "Line 71: CAM5 should be spelled out as "Community Atmosphere Model version 5"*

**Response:** Accepted. We reword this sentence. Please see Line 88.

*Line 80: Change "simulation" to "simulating"*

**Response:** Accepted. Please see Line 90.

*Line 108: How about global emissions for 2016? No changes in emissions of other regions? The CEDS emissions only went up to 2014.*

**Response:** Accepted. The global emissions other than China used in all simulations are from the CEDS for 2008, because we focus on impacts of the China's emission variations on aerosol radiative effects. We add more description for emissions. Please see Section 2.2.

*Line 127-128: Why MISR AOD is used here rather than other remote sensing or hybrid products that are often used by the aerosol modeling community to evaluate models.*

**Response:** Accepted. Because the inter-annual variations in observed AOD over China in the recent decade are consistent between MISR and other remote sensing products, such as MODIS (shown in J. Li, 2020), only MISR AOD was used in the original manuscript. We add the spatial maps of MODIS AOD for 2008 and 2016. The detailed values of observations and the comparison with simulations have been given in the main text and supplementary figures. Please see Figure S4 and Figure S5 and Line 211-221.

*Line 164-165: this sentence needs clarification. Do you mean less ammonium sulfate leads to more ammonium nitrate over East China in 2016? Assuming that all else being equal?*

**Response:** Accepted. In the case with only $SO_2$ emission variation included (Exp16SO2), nitrate concentrations increase by almost the same level with those in the 2016 case (Exp16, with changes in all emission variables). Therefore, the reductions in sulfate have led to elevated particulate nitrate concentrations during 2008–2016, given that $NH_3$ emissions are unchanged. This is caused by the aerosol thermodynamic equilibrium in the sulfate-nitrate-ammonium-water system (Chapter 10 in Seinfeld and Pandis, 2016). We add more sentences to explain the increase of nitrate concentrations resulted from the reductions in sulfate. Please see Line 191-199.

*Figure 4: It's very difficult to see the AERONET observations in the figure. Is there only one site that can represent East Asia? I don't see much value added here.*

**Response:** Accepted. We modify Figure 4 to show the location of the AERONET station more clearly. There are some AERONET stations located in China, but only Beijing and Xianghe stations have long-term data available from 2008 to 2016 (also used by Li, 2020). Because these two sites correspond to the same model horizontal grid, we use the averaged value from them to show the long-term variation of AOD for the region. Please see Line 239-243 and Figure 4.

*Figure 6: How was the total RF calculated? How does this compared to the TOA forcing difference between Exp08 and Exp16?*

**Response:** Accepted. The total RF induced by the aerosols is calculated by combining the aerosol DRF and CRF at TOA derived from the Exp08 and Exp16 cases. The CRF has been scaled based on the IPCC AR5 report. Please see Line 289-292 and the Figure 7 caption. We understand that the phrase "TOA forcing difference" mentioned by the referee here means the regional-mean change in the net radiation flux at TOA between the two cases. It is estimated to be +0.46 W m$^{-2}$, quite close to the sum of aerosol DRF and CRF

($+0.50$ W m$^{-2}$). The CRF mentioned here is the original estimate by the model without a scaling.

References:

Boucher, O., D. Randall, P. Artaxo, C. Bretherton, G. Feingold, P. Forster, V.-M. Kerminen, Y. Kondo, H. Liao, U. Lohmann, P. Rasch, S.K. Satheesh, S. Sherwood, B. Stevens and X.Y. Zhang, 2013: Clouds and Aerosols. In: Climate Change 2013: The Physical Science Basis. Contribution of Working Group I to the Fifth Assessment Report of the Intergovernmental Panel on Climate Change [Stocker, T.F., D. Qin, G.-K. Plattner, M. Tignor, S.K. Allen, J. Boschung, A. Nauels, Y. Xia, V. Bex and P.M. Midgley (eds.)]. Cambridge University Press, Cambridge, United Kingdom and New York, NY, USA.

Fadnavis, S., Müller, R., Kalita, G., Rowlinson, M., Rap, A., Li, J.-L. F., Gasparini, B., and Laakso, A.: The impact of recent changes in Asian anthropogenic emissions of SO2 on sulfate loading in the upper troposphere and lower stratosphere and the associated radiative changes, Atmos. Chem. Phys., 19, 9989-10008, 10.5194/acp-19-9989-2019, 2019.

Ghan, S. J.: Technical Note: Estimating aerosol effects on cloud radiative forcing, Atmos. Chem. Phys., 13, 9971-9974, 10.5194/acp-13-9971-2013, 2013.

Ghan, S., Wang, M., Zhang, S., Ferrachat, S., Gettelman, A., Griesfeller, J., Kipling, Z., Lohmann, U., Morrison, H., Neubauer, D., Partridge, D. G., Stier, P., Takemura, T., Wang, H., and Zhang, K.: Challenges in constraining anthropogenic aerosol effects on cloud radiative forcing using present-day spatiotemporal variability, Proceedings of the National Academy of Sciences, 113, 5804, 10.1073/pnas.1514036113, 2016.

He, J., Zhang, Y., Glotfelty, T., He, R., Bennartz, R., Rausch, J., and Sartelet, K.: Decadal simulation and comprehensive evaluation of CESM/CAM5.1 with advanced chemistry, aerosol microphysics, and aerosol-cloud interactions, J. Adv. Model. Earth Syst., 7, 110-141, 10.1002/2014ms000360, 2015.

Liu, M., et al.: Trends of Precipitation Acidification and Determining Factors in China During 2006–2015, J. Geophys. Res.-Atmos, 125, e2019JD031301, 10.1029/2019JD031301, 2020.

Li, C., McLinden, C., Fioletov, V., Krotkov, N., Carn, S., Joiner, J., Streets, D., He, H., Ren, X., Li, Z., and Dickerson, R. R.: India Is Overtaking China as the World's Largest Emitter of Anthropogenic Sulfur Dioxide, Sci Rep, 7, 14304, 10.1038/s41598-017-14639-8, 2017.

Li, J.: Pollution Trends in China from 2000 to 2017: A Multi-Sensor View from Space, Remote Sensing, 12, 10.3390/rs12020208, 2020.

Matsui, H.: Development of a global aerosol model using a two-dimensional sectional method: 1. Model design, J. Adv. Model. Earth Syst., 9, 1921-1947, 10.1002/2017ms000936, 2017.

Matsui, H., and Mahowald, N.: Development of a global aerosol model using a two-dimensional sectional method: 2. Evaluation and sensitivity simulations, J. Adv. Model. Earth Syst., 9, 1887-1920, 10.1002/2017ms000937, 2017.

Samset, B. H., Lund, M. T., Bollasina, M., Myhre, G., and Wilcox, L.: Emerging Asian aerosol patterns, Nat. Geosci., 12, 582-584, 10.1038/s41561-019-0424-5, 2019.

Schwarz, M., Folini, D., Yang, S., Allan, R. P., and Wild, M.: Changes in atmospheric shortwave absorption as important driver of dimming and brightening, Nat. Geosci., 13, 110-115, 10.1038/s41561-019-0528-y, 2020.

Seinfeld, J. H., and Pandis, S. N.: Atmospheric Chemistry and Physics: From Air Pollution to Climate Change, Third edition ed., John Wiley & Sons, Inc., 2016.

Sockol, A., and Small Griswold, J. D.: Intercomparison between CMIP5 model and MODIS satellite-retrieved data of aerosol optical depth, cloud fraction, and cloud-aerosol interactions, Earth Space Sci., 4, 485-505, 10.1002/2017ea000288, 2017.

Zhang, Q., et al.: Drivers of improved PM2.5 air quality in China from 2013 to 2017, Proceedings of the National Academy of Sciences, 116, 24463, 10.1073/pnas.1907956116, 2019.

Zhang, X., et al.: The impact of meteorological changes from 2013 to 2017 on PM2.5 mass reduction in key regions in China, Science China Earth Sciences, 62, 1885-1902, 10.1007/s11430-019-9343-3, 2019.

---

## Author Comment (AC2) · 27 Oct 2020

**Response to Referee #2**

Thanks very much for the Referee #2's helpful comments. We revise our manuscript according to the annotated document. Please see the point-to-point responses in the following.

*Referee: "I recommend a more focused title, rather than a whole sentence."*

**Response:** Accepted. The title has been changed to "Aerosol radiative forcings induced by variations in anthropogenic emissions over China during 2008–2016".

*Referee: "result->resulted; latest->the latest"*

**Response:** Accepted. We reword them. Please see Line 6-10 in the revised manuscript.

*Referee: "Line 25: This phrase is a non-sequitor with respect to what is given above. Is an aerosol-induced positive radiative forcing truly desirable?"*

**Response:** Accepted. Because aerosol radiative forcings induced by anthropogenic emissions potentially influence surface temperature and precipitation, we imply that targeted regulations of emissions are required to mitigate the risk of climate change. We reword the sentence to highlight implications of this study. Please see Line 26-28.

*Referee: "Line 60: the->a; don't->do not"*

**Response:** Accepted. We modify them. Please see Line 67-68.

*Referee: "Line 72: A verb is missing from this phrase."*

**Response:** Accepted. We reword the sentence. Please see Line 90.

*Referee: "Line 88: ... species for years 2008 and 2016 for...."*

**Response:** Accepted. We reword the sentence. Please see Line 105.

*Referee: "Line 91: Also for all the 2016 simulations? what is the reasoning behind that? unless you used the proper 2016 meteorology for the 2016 simulations and the phrase is simply misleading. Line 96: As long as you used the 2008 meteo for the 2008 run and the 2016 meteo for the 2016 run. Otherwise, you cannot separate anything, I think."*

**Response:** Because change in meteorological conditions (clouds, winds, etc.) could also influence the calculation of aerosol radiative effects, we need to separate the contribution of emission variations on aerosol radiative effects by using the fixed meteorological fields from one fixed year. Such kind of separation is commonly adopted in previous studies on evaluating aerosol climate effects due to emission variations for a long-term period (e.g., Leibensperger et al., 2012; Fadnavis et al., 2019).

We add more description for the treatment of meteorological fields in this study and perform a sensitivity simulation to discuss the effects of meteorological conditions on aerosol radiative forcings. Please see Line 113-115, Line 338-346, and Table S4.

*Referee: "Table 2: I think you need to define with lat/lon boxes what EC is and also you need to add the reference to the emission inventory in the table legend."*

**Response:** Accepted. We add the definition of EC and the reference to the emission. Please see the Table 2 caption.

*Referee: "Line 123: Is this data public? give a website and a reference to this data. Have they been used before in studies? have they been validated? you should make this clear before you use them to validate your model runs."*

**Response:** Accepted. The EANET databases have been widely used for air pollution studies and model evaluation of inorganic aerosol components (e.g., Itahashi et al., 2018). The program has adopted the regular Quality Assurance/Quality Control. Details can be

found at: https://www.eanet.asia/. We add more description for the database. Please see Line 140-144 and Data availability.

*Referee: "Line 123: including->include."*

**Response:** Accepted. Please see Line 143.

*Referee: "Line 123: From the text above the impression given is that you run your model only for the Chinese domain. Do you mean you also run for Japan? I think a plot showing the model domain with a species of your choice would be a nice addition to your section 2.1. You should also address the issue whether the chosen emission inventory, species concentrations, etc. that apply to China also apply to Japan."*

**Response:** Accepted. The climate model, CAM5/ATRAS, was ran on a global scale in this study. The model configuration is consistent among different regions. The emission inventory used for China is from MEIC and the rest of the world from CMIP6. Please see Line 103-107 and Section 2.2.

*Referee: "Line 128: Which version of the data did you use? downloaded from where? did you regrid? you need to provide references for this data and discuss their validation. How do they compare for e.g. against MODIS, which is the more established satellite dataset?"*

**Response:** Accepted. The version and download source of MISR data have been included in the revised manuscript. Further, we add the MODIS AOD data for comparison with the model. The differences between modeled and observed AOD from MISR and MODIS are shown in the Supplementary Materials. The detailed comparison between MISR and MODIS has been provided by Garay et al. (2020) and references therein. Please see Line 155-159 and Figure S4 and S5.

*Referee: "Line 132: This agreement is not shown by Figure 1a/b. You need a different representation, either a line plot, since your stations are point locations or a table with numbers.*

*You also have to be very clear as to how you handdled the ground-based measurements. You are comparing a point observation from the ground with a model with moderately big pixels. How did you deal with representativeness errors? SO2 has a very clear seasonal cycle, how does that apply to your comparisons? do you create monthly mean values from the observations? for which time of the day? There are many choices that can affect your comparisons and those need to be made clear in the text, in Section 2.3."*

**Response:** Accepted. We demonstrate that our model can represent the inter-annual variations in aerosol concentrations on a regional scale for 2008–2016 by comparing with available ground-based and satellite observations as well as previous studies.

1) We add the quantitative comparison between modeled and observed sulfate concentrations (Table S2). Please see Line 162-176, Table S1, and Table S2.

2) More details on the use of ground-based observation data are added in the methods section and supplementary materials. Because monthly observation data for aerosol components are very limited and not publicly available in China (particularly for 2008), we average the data derived from different measurement campaigns and compare them with the annual-mean simulation results extracted from the model horizontal grid closest to the observation sites. Please see Line 140-147, Table S1, and Table S2.

3) Although monthly observations of sulfate are not available, we validate the model performance of $SO_2$ seasonality using satellite-based $SO_2$ column measurements by OMI. The model presents a consistent seasonal pattern of $SO_2$ concentrations with satellite retrievals in China. Please see Figure S2.

4) It's clear that both the observations and simulations have demonstrated notable decreases of sulfate concentrations over East China, as shown in Figure 1 and Table S2. Representation errors of in-situ observations are possible, but they could not change the temporal variations of sulfate concentrations, which are highly dependent on regional emissions in China (Zheng et al., 2015).

*Referee: "Line 135: Again, this agreement is not demonstrated and the findings of Liu et al. should be stated here in numbers."*

**Response:** Accepted. We add the detailed result given by Liu et al. (2018). Please see Line 171-172.

*Referee: "Line 138: Again, you need to explicitly give the findings of Li et al. that actually agree with your 10microgram m-3 finding."*

**Response:** Accepted. We show the findings of Li et al (2017) to support our results. Please see Line 166-169.

*Referee: "Line 141: What percentage of the wind is downwind per annum?"*

**Response:** Accepted. The long-range transport of aerosols in China is dominated by the westerly winds, which are prevailing winds from the west toward the east in the middle latitudes between 30 °N and 60 °N (Liang et al., 2004). Because our simulations were conducted on a global scale, the quantitative impacts of China's emission variations on the aerosol distributions in different downwind regions can be seen clearly in Fig. 2. We improve the sentence. Please see Line 175-176.

*Referee: "Line 142: verity->verify."*

**Response:** Accepted. We modify this word. Please see Line 177.

*Referee: "Line 146: You cannot find a trend by two years only. If you had all the years between 2008 and 2016, i.e. 9 years, then you might be able to claim this. Your simulations show a difference between two years, this is not a trend."*

**Response:** Accepted. The results mentioned here are derived from the simulations in 2008 and 2016. Nevertheless, the variations of BC and sulfate in the model are in line with exiting observations, which have demonstrated the decreasing trends over the past decade (Liu et al., 2018; Kanaya et al., 2020). We reword the sentence. Please see Line 177-183.

*Referee: "Line 147: You need to mark this island for those of us who do not know where it is and show in a figure this decline. Again, if you are showing results from two years only this is not an "interannual" decline, this is a decline between two years."*

**Response:** Accepted. The decline of BC concentrations in Fukue island between 2008 and 2016 has been noted in Figure 1c, d. The long-term observations in Fukue island show a significantly decreasing trend in BC concentrations (Kanaya et al., 2020) and support our simulation results.

*Referee: "Line 150: This fact was not made clear in a satisfactory manner to the audience. You need more precise statistics."*

**Response:** Accepted. As suggested by the referee, we add more detailed statistical data for the comparison between simulations and observations as well as previous studies. We also reword this sentence. Please see Table S3, Line 162-190.

*Referee: "Figure 1: You mean, the locations of the EANET stations are shown, not the actual numbers these provide. Replace literatures with literature."*

**Response:** Accepted. We replace the word literature used here. The observed aerosol concentrations (the values) at EANET stations are shown in Figure 1. The inner colors of each circle represent the magnitude of aerosol concentrations, which are overlaid on the map of simulation results. The detailed location information of EANET stations is given in Table S1.

*Referee: "Figure 2: This is not a map of China, this is a global map. Does this mean you run your simulations for the entire globe? this should be made clear in Section 2.1."*

**Response:** Accepted. We run the global climate model on a global scale. Please see Line 103-104.

*Referee: "Line 162: Not shown clearly. Impossible to tell what colour the dots have."*

**Response:** Accepted. The comparison of simulated nitrate concentrations with observations can be seen in both Table S2 and Figure S3. The observed concentrations are shown in colored dots, which are overlaid on the map of simulations.

*Referee: "Line 165: Similar, i.e.? in numbers?"*

**Response:** Accepted. The numbers are added here. Please see Line 196-198 in the revised manuscript.

*Referee: "Line 182: Not a trend if only two years are used."*

**Response:** Accepted. We reword the phrase in this section. Please see Line 211.

*Referee: "Line 184: This is a very important finding and should be included in the main paper."*

**Response:** Thanks for the suggestion. Because similar findings for the comparison between satellite-observed and modeled AOD have been reported in several previous studies (He et al., 2015; Sockol and Small Griswold, 2017), we only display those results in the Supplementary Materials (Figure S4 and S5).

*Referee: "Line 187: This can be due to a million different reasons. You do not give an std, maybe the respective errors compensate for this difference. Also, how exactly did you calculate the annual mean AOD from the two sets of information? did you collocate on a daily basis, then created the monthly and then the annual? another way? these simple choices may affect your numbers greatly."*

**Response:** Accepted. We show the standard deviations (i.e., uncertainties ranges given by the MISR product, please see Garay et al., 2020) and inter-annual trends in AOD values from MISR. The MISR aerosol product at a one-year temporal resolution is used in this study. Please see Figure S6 and Line 155-159 and Line 227-228 in the revised manuscript.

[Figure]

Fig. S6 Inter-annual variation in AOD at 550 nm retrieved by MISR over East China during 2008–2016. The mean values (dots) and standard deviations (vertical lines) are calculated using the gridded AOD and standard deviation from the Level-3 MISR product with a one-year temporal resolution. The black line represents the linear fit of the AOD (T-test: α=0.013<0.05).

*Referee: "Line 188: How negative? precisely."*

**Response:** Accepted. The numbers are added in the statement. Please see Line 218-220.

*Referee: "Figure 4: So, these are the global changes due to the local emission variation? which experiment is this, from Table1? the first?"*

**Response:** Accepted. The changes are derived from the Exp08 and Exp16 simulations. All the experiments are performed on a global scale but only results in China and outflow areas used for analysis. Figure 4 shows the changes in East Asia and continental outflows. Please see Line 103-115.

*Referee: "Figure 4: Where are the coloured dots?"*

**Response:** Accepted. We update Figure 4 to show the location of the AERONET station. The detailed values of observations and the comparison with simulations have been given in the main text. Please see Line 237-242.

*Referee: "Line 206: That may be the case, but from the material you provide, it cannot be verified."*

**Response:** Accepted. More details on the model evaluation on aerosol optical properties and associated uncertainties are added in the section. We reword this sentence here based on the evidence we provided. Please see Line 222-246.

*Referee: "Line 244: You need to show these results."*

**Response:** Accepted. The results have been shown in this sentence. Please see Line 285-287.

*Referee: "Figure 5: Why not zoom these maps to show Asia only? more detail would appear and it is clear that the rest of the world is not affected."*

**Response:** Accepted. We zoom out the Figure to underline the variations in Asia and surrounding areas. Please see Figure 5 and Figure 6 in the revised manuscript.

*Referee: "Figure 5f: This map is impossibly noisy, neighboring pixels show -1 and 1 Wm-2 forcing, this is not physical."*

**Response:** We would like to point out that the response of cloud radiative forcing to aerosol perturbations is distinguishable from the noise over China and north Pacific (Figure 5f). The aerosol-induced cloud radiative forcing (CRF) is estimated by varying the anthropogenic emissions between 2008 and 2016 in two different simulations. The CRF noise can be caused by the aerosol-meteorology feedback in the global climate model. Such anomalies of aerosol-induced CRF signals are also seen in similar studies (e.g., Shi and Liu, 2019).

References:

Fadnavis, S., Müller, R., Kalita, G., Rowlinson, M., Rap, A., Li, J.-L. F., Gasparini, B., and Laakso, A.: The impact of recent changes in Asian anthropogenic emissions of SO2 on sulfate loading in the upper troposphere and lower stratosphere and the associated radiative changes, Atmos. Chem. Phys., 19, 9989-10008, 10.5194/acp-19-9989-2019, 2019.

Garay, M. J., Witek, M. L., Kahn, R. A., Seidel, F. C., Limbacher, J. A., Bull, M. A., Diner, D. J., Hansen, E. G., Kalashnikova, O. V., Lee, H., Nastan, A. M., and Yu, Y.: Introducing the 4.4 km spatial resolution Multi-Angle Imaging SpectroRadiometer (MISR) aerosol product, Atmos. Meas. Tech., 13, 593-628, 10.5194/amt-13-593-2020, 2020.

He, J., Zhang, Y., Glotfelty, T., He, R., Bennartz, R., Rausch, J., and Sartelet, K.: Decadal simulation and comprehensive evaluation of CESM/CAM5.1 with advanced chemistry, aerosol microphysics, and aerosol-cloud interactions, J. Adv. Model. Earth Syst., 7, 110-141, 10.1002/2014ms000360, 2015.

Itahashi, S., Yumimoto, K., Uno, I., Hayami, H., Fujita, S. I., Pan, Y., and Wang, Y.: A 15-year record (2001–2015) of the ratio of nitrate to non-sea-salt sulfate in precipitation over East Asia, Atmos. Chem. Phys., 18, 2835-2852, 10.5194/acp-18-2835-2018, 2018.

Kanaya, Y., Yamaji, K., Miyakawa, T., Taketani, F., Zhu, C., Choi, Y., Komazaki, Y., Ikeda, K., Kondo, Y., and Klimont, Z.: Rapid reduction in black carbon emissions from China: evidence from 2009–2019 observations on Fukue Island, Japan, Atmos. Chem. Phys., 20, 6339-6356, 10.5194/acp-20-6339-2020, 2020.

Leibensperger, E. M., Mickley, L. J., Jacob, D. J., Chen, W.-T., Seinfeld, J. H., Nenes, A., Adams, P. J., Streets, D. G., Kumar, N., and Rind, D.: Climatic effects of 1950–2050 changes in US anthropogenic aerosols – Part 1: Aerosol trends and radiative forcing, Atmos. Chem. Phys., 12, 3333–3348, https://doi.org/10.5194/acp-12-3333-2012, 2012.

Liu, M., Huang, X., Song, Y., Xu, T., Wang, S., Wu, Z., Hu, M., Zhang, L., Zhang, Q., Pan, Y., Liu, X., and Zhu, T.: Rapid SO2 emission reductions significantly increase tropospheric ammonia concentrations over the North China Plain, Atmos. Chem. Phys., 18, 17933-17943, 10.5194/acp-18-17933-2018, 2018.

Li, C., McLinden, C., Fioletov, V., Krotkov, N., Carn, S., Joiner, J., Streets, D., He, H., Ren, X., Li, Z., and Dickerson, R. R.: India Is Overtaking China as the World's Largest Emitter of Anthropogenic Sulfur Dioxide, Sci Rep, 7, 14304, 10.1038/s41598-017-14639-8, 2017.

Liang, Q., Jaeglé, L., Jaffe, D. A., Weiss-Penzias, P., Heckman, A., and Snow, J. A.: Long-range transport of Asian pollution to the northeast Pacific: Seasonal variations and transport pathways of carbon monoxide, J. Geophys. Res.-Atmos, 109, 10.1029/2003JD004402, 2004.

Shi, Y., and Liu, X.: Dust Radiative Effects on Climate by Glaciating Mixed-Phase Clouds, Geophys. Res. Lett., 46, 6128-6137, 10.1029/2019gl082504, 2019.

Sockol, A., and Small Griswold, J. D.: Intercomparison between CMIP5 model and MODIS satellite-retrieved data of aerosol optical depth, cloud fraction, and cloud-aerosol interactions, Earth Space Sci., 4, 485-505, 10.1002/2017ea000288, 2017.

Zheng, G. J., Duan, F. K., Su, H., Ma, Y. L., Cheng, Y., Zheng, B., Zhang, Q., Huang, T., Kimoto, T., Chang, D., Pöschl, U., Cheng, Y. F., and He, K. B.: Exploring the severe winter haze in Beijing: the impact of synoptic weather, regional transport and heterogeneous reactions, Atmos. Chem. Phys., 15, 2969–2983, https://doi.org/10.5194/acp-15-2969-2015, 2015.

---

## Author Comment (AC3) · 27 Oct 2020

Dear Editor, Please see our responses to the referees' comments. Thanks very much for your help. Best wishes, Mingxu Liu

---

## Author Response (AR1)

**Response to Referee #1**

We sincerely appreciate the Referee #1's comments, which indeed help use to improve the manuscript. We accept all the suggestions proposed by the Referee and revise the manuscript accordingly. Please see the point-point responses to the comments in the following.

*Referee: "One global model is used to study the impact of changes in anthropogenic aerosol emissions in China on regional radiation budget. The authors found that, compared to 2008, reductions in black carbon and sulfur dioxide emissions in 2016, separately, had an opposite effect on radiative forcing, resulting in a net radiative forcing of -0.04 W/m2 by the changes in black carbon and sulfur dioxide emissions. The main scientific conclusion of this study is by no means novel. A number of studies in the literature have shown similar conclusions over China, East Asia and/or other regions around the world. In addition, I have several major specific concerns regarding the methodology and model biases as described below"*

**Response:** Accepted. We improve the description of the novelty and the scientific implications of this study in the new text. The novelty mainly includes:

1) The clean-air measures in China led to the heaviest-ever reductions in anthropogenic emissions over the past decade (please see Table 1 in our manuscript), which provides a unique opportunity to investigate the aerosol radiative forcings due to such substantial variations in emissions. We quantified the aerosol-induced radiative forcings at TOA due to both aerosol-radiation interactions and aerosol-cloud interactions over China and the outflow regions (North Pacific) using a global climate model and the latest China's anthropogenic emission inventory (replacing CMIP6 emissions for China). To our knowledge, no modeling studies have shown such results before. Please see Line 60-77 in the revised manuscript (Unless noted otherwise, line numbers are shown for the revised manuscript).

Our results will implicate for predicting climate impacts of emission variations under different SSP (shared socio-economic pathways) scenarios in the future (e.g., Samset et al., 2019). Please see Section 3.3 (Line 320-337).

2) Recent studies mainly emphasized the radiative forcings of $SO_2$ emission variations over China in the past decade (Fadnavis et al., 2019), but didn't uncover their interactions with BC and other aerosol species (e.g., nitrate) in determining the net aerosol radiative forcings. Our results suggest that the reductions in $SO_2$ emissions and the associated sulfate concentrations would facilitate the formation of ammonium nitrate and also diminish the radiative forcing of BC-containing particles through the absorption enhancement effect. These complex effects should be considered in the assessment of aerosol radiative forcings. Please see Line 257-264.

3) This study indicates that the reductions in sulfate and BC led to the increase in downward solar radiation flux onto the Earth's surface. The results provide the first modeling evidence for the cause of surface brightening in China over the past decade, as shown by the long-term observations (Schwarz et al., 2020). Please see Line 301-319.

*Major comments*

*Referee: "1. "the recent decade" in the title and abstract (and a few other places in the main text) should be changed to the exact time period "2008-2016". Why are these two specific years chosen? Many published studies on aerosol radiative forcing use 2014 as "present day". On the other hand, the simulations were only performed for two years with one year discarded and the results were mostly presented as the difference between the two time slices. I don't think the use "trend" in the description of discussion of results is appropriate."*

**Response:** Accepted. We use "2008–2016" in the revised title.

We chose these two years for analysis because the differences of emissions between these two years (2008 and 2016) can reflect the decadal changes in China's anthropogenic emissions caused by a series of emission control policies from 2008 to the present. The $SO_2$ emissions in China peaked around 2007 and have declined since then (Li et al., 2017). And the toughest-ever clean-air policies were conducted between 2013–2017 and led to further reductions in primary aerosol emissions (Zhang et al., 2019). We added more description on the study period in the revised manuscript. Please see Line 79-83.

We revise the statement with the use "trend" in the main text. Some trend results from previous studies and long-term observation data have been shown to support our simulation results. Please see Line 162-183 and Fig. S6.

*Referee: "Given the CAM experiment design, comparison of model results in 2016, which is based on meteorological conditions in 2008, to observations is not apples-to-apples. Meteorology is probably quite important for interannual variability in aerosol concentrations (e.g., Yang et al., 2018). Please discuss how this setup could affect your conclusions."*

**Response:** Accepted. We performed another simulation using both the emissions and meteorology fields for 2016 and discussed how meteorology conditions affect aerosol concentrations and the associated radiative effects. Please see Line 338-346 and Table S4 in the revised manuscript and SI.

Both this study and previous studies demonstrate that the substantial reductions in anthropogenic emissions dominated the mitigation of aerosol pollutions in China in the past decade (Liu et al., 2020; Zhang Q. et al., 2019; Zhang X., et al., 2019). Variation in meteorological conditions only has a minor effect on the decadal variability in aerosol concentrations and their radiative effects due to aerosol-radiation interactions over China (Table S2).

Note that the aerosol-induced cloud radiative forcing cannot be separated from the changes of meteorological fields. This study mainly focuses on the impacts of inter-annual variations in anthropogenic emissions on aerosol radiative effects and meteorological factors are not the key topic of this study.

*Referee: "2. I also have a concern about the current experimental design and forcing estimates, which cannot be used to support one of the main conclusion (i.e., the net BC and sulfate radiative forcing of -0.04 W m-2). Forcing induced by BC emissions reduction and forcing induced by SO2 reduction are diagnosed in separate simulations with one of the individual aerosol changes included, assuming that the*

*forcings induced by the individual aerosol changes are additive. Results in the current study and some previous studies (e.g., Chen et al., 2017) have shown nonlinearities between emission and forcing changes even for the same species."*

**Response:** In this study, the forcings induced by BC and $SO_2$ emissions were calculated online in the same simulations (i.e., differences between Exp16 and Exp08). The separate simulations (mentioned by the referee) with one of the emission change included were only used for comparison instead of drawing the main conclusion. Therefore, the nonlinearities between emissions and forcings have been considered in our study. We add more description for the calculation of aerosol forcings. Please see Line 103-115.

*Referee: "On the other hand, the CRF in this study is calculated by scaling results of different simulations with the best estimate of effective radiative forcing in IPCC AR5, which needs to be described more clearly. I am not sure how exactly this was done and why this number is comparable to the DRF calculated in this study. As far as I know, the AR5 effective radiative forcing is calculated differently from the method of Ghan (2013). The latter attributes radiation changes induced by above-cloud light absorbing aerosols to DRF."*

**Response:** Accepted. In this study, we scaled the global-mean CRF (aerosol effects on cloud forcings) calculated in our model to the best estimate of the effective radiative forcing due to aerosol-cloud interactions ($-0.5$ W m$^{-2}$) given in Chapter 7 of IPCC AR5 report (Boucher et al., 2013), which makes our CRF results consistent with the current best estimate of CRF. We do not scale the DRF estimate in this study. The DRF is online diagnosed in the model using the method of Ghan (2013) that attributes the above-cloud light absorbing effects by aerosols to DRF. We reword the description for the calculation of CRF in this study. Please see Line 116-126.

*Referee: "3. It looks to me that the CAM AOD is so much off comparing to MISR (Figure S2), especially over the continental outflow regions and the high latitudes. A difference plot would be able to show the biases more clearly. There are various studies in the literature that have shown improved aerosol simulations in CAM5 (e.g., Ghan et al. 2016 and references therein). Why aren't those improvements included in this version of the CAM model? Model biases in AOD as well as the vertical distribution of aerosols and how they can impact the forcing estimates in the two regions of interest need to be quantitatively evaluated and more objectively discussed."*

**Response:** Accepted. We add more discussions on the model bias in AOD, compare the vertical distribution of modeled BC concentrations with aircraft measurements, and evaluate the uncertainties in the forcing estimates. Here, we point out that:

1) A difference plot between modeled and observed AOD from MISR and MODIS have been given in the Supplementary Materials (Figure S5). The model biases in AOD over some continental outflow regions and high latitudes (mainly in Southeastern Asia and East of Africa) are commonly found in CAM models (Sockol and Griswold, 2018; He et al., 2015), even if improvements for aerosol simulations have been included. It's quite difficult to quantitatively evaluate the model biases of radiative forcing estimates, but a qualitatively description of the potential bias related to AOD has been added in the revised manuscript. Please see Line 225-236 and Line 274-279.

2) The aerosol simulations have been markedly improved by our group based on standard CAM models (Matsui 2017; Matsui and Mahowald, 2017). Our model uses an advanced two-sectional bin representation of size and mixing states of BC particles, which calculates detailed aerosol processes and their interactions with radiation and clouds. We also include a new pathway of secondary sulfate formation as well as an improved volatility basis-set approach for secondary organic aerosols. Our simulations can generally reproduce the spatiotemporal features of major aerosol compositions over China and outflow areas (Figure 2, Figure S3, and Table S2). In our understanding, the paper mentioned by the referee, i.e., Ghan et al. (2016), does not discus improvements of aerosol simulations.

3) Because the observations of vertical profiles of sulfate and BC for a long period (> 1 month) are quite limited in China mainland, we evaluate the model performances of BC profiles in the continental outflow region (i.e., North Pacific) using the aircraft measurements from the HIPPO campaign. Please see Figure S1 and Line 183-186 in the revised manuscript.

*Minor comments and technical edits:*

*Referee: "Line 57-59: I believe there are quite a few other studies in the literature that are relevant here."*

**Response:** Accepted. We add relevant literatures in the revised manuscript. To our knowledge, very few studies have focused on the aerosol radiative forcings in response to the substantial emission variations in China over the past decade. Please see Line 61-72.

*Referee: "Line 71: CAM5 should be spelled out as "Community Atmosphere Model version 5"*

**Response:** Accepted. We reword this sentence. Please see Line 88.

*Line 80: Change "simulation" to "simulating"*

**Response:** Accepted. Please see Line 90.

*Line 108: How about global emissions for 2016? No changes in emissions of other regions? The CEDS emissions only went up to 2014.*

**Response:** Accepted. The global emissions other than China used in all simulations are from the CEDS for 2008, because we focus on impacts of the China's emission variations on aerosol radiative effects. We add more description for emissions. Please see Section 2.2.

*Line 127-128: Why MISR AOD is used here rather than other remote sensing or hybrid products that are often used by the aerosol modeling community to evaluate models.*

**Response:** Accepted. Because the inter-annual variations in observed AOD over China in the recent decade are consistent between MISR and other remote sensing products, such as MODIS (shown in J. Li, 2020), only MISR AOD was used in the original manuscript. We add the spatial maps of MODIS AOD for 2008 and 2016. The detailed values of observations and the comparison with simulations have been given in the main text and supplementary figures. Please see Figure S4 and Figure S5 and Line 211-221.

*Line 164-165: this sentence needs clarification. Do you mean less ammonium sulfate leads to more ammonium nitrate over East China in 2016? Assuming that all else being equal?*

**Response:** Accepted. In the case with only $SO_2$ emission variation included (Exp16SO2), nitrate concentrations increase by almost the same level with those in the 2016 case (Exp16, with changes in all emission variables). Therefore, the reductions in sulfate have led to elevated particulate nitrate concentrations during 2008–2016, given that $NH_3$ emissions are unchanged. This is caused by the aerosol thermodynamic equilibrium in the sulfate-nitrate-ammonium-water system (Chapter 10 in Seinfeld and Pandis, 2016). We add more sentences to explain the increase of nitrate concentrations resulted from the reductions in sulfate. Please see Line 191-199.

*Figure 4: It's very difficult to see the AERONET observations in the figure. Is there only one site that can represent East Asia? I don't see much value added here.*

**Response:** Accepted. We modify Figure 4 to show the location of the AERONET station more clearly. There are some AERONET stations located in China, but only Beijing and Xianghe stations have long-term data available from 2008 to 2016 (also used by Li, 2020). Because these two

sites correspond to the same model horizontal grid, we use the averaged value from them to show the long-term variation of AOD for the region. Please see Line 239-243 and Figure 4.

*Figure 6: How was the total RF calculated? How does this compared to the TOA forcing difference between Exp08 and Exp16?*

**Response:** Accepted. The total RF induced by the aerosols is calculated by combining the aerosol DRF and CRF at TOA derived from the Exp08 and Exp16 cases. The CRF has been scaled based on the IPCC AR5 report. Please see Line 289-292 and the Figure 7 caption.

We understand that the phrase "TOA forcing difference" mentioned by the referee here means the regional-mean change in the net radiation flux at TOA between the two cases. It is estimated to be +0.46 W m$^{-2}$, quite close to the sum of aerosol DRF and CRF (+0.50 W m$^{-2}$). The CRF mentioned here is the original estimate by the model without a scaling.

*Referee: "Line 128: Which version of the data did you use? downloaded from where? did you regrid? you need to provide references for this data and discuss their validation. How do they compare for e.g. against MODIS, which is the more established satellite dataset?"*

**Response:** Accepted. The version and download source of MISR data have been included in the revised

manuscript. Further, we add the MODIS AOD data for comparison with the model. The

differences between modeled and observed AOD from MISR and MODIS are shown in the

Supplementary Materials. The detailed comparison between MISR and MODIS has been

provided by Garay et al. (2020) and references therein. Please see Line 155-159 and Figure

S4 and S5.

*Referee: "Line 132: This agreement is not shown by Figure 1a/b. You need a different representation,*
*either a line plot, since your stations are point locations or a table with numbers. You also have to be*
*very clear as to how you handdled the ground-based measurements. You are comparing a point*
*observation from the ground with a model with moderately big pixels. How did you deal with*
*representativeness errors? SO2 has a very clear seasonal cycle, how does that apply to your comparisons?*
*do you create monthly mean values from the observations? for which time of the day? There are many*
*choices that can affect your comparisons and those need to be made clear in the text, in Section 2.3."*

**Response:** Accepted. We demonstrate that our model can represent the inter-annual variations in aerosol

concentrations on a regional scale for 2008–2016 by comparing with available ground-based

and satellite observations as well as previous studies.

1) We add the quantitative comparison between modeled and observed sulfate concentrations

(Table S2). Please see Line 162-176, Table S1, and Table S2.

2) More details on the use of ground-based observation data are added in the methods section

and supplementary materials. Because monthly observation data for aerosol components are

very limited and not publicly available in China (particularly for 2008), we average the data

derived from different measurement campaigns and compare them with the annual-mean

simulation results extracted from the model horizontal grid closest to the observation sites.

Please see Line 140-147, Table S1, and Table S2.

3) Although monthly observations of sulfate are not available, we validate the model

performance of $SO_2$ seasonality using satellite-based $SO_2$ column measurements by OMI. The

model presents a consistent seasonal pattern of $SO_2$ concentrations with satellite retrievals in China. Please see Figure S2.

4) It's clear that both the observations and simulations have demonstrated notable decreases of sulfate concentrations over East China, as shown in Figure 1 and Table S2. Representation errors of in-situ observations are possible, but they could not change the temporal variations of sulfate concentrations, which are highly dependent on regional emissions in China (Zheng et al., 2015).

*Referee: "Line 135: Again, this agreement is not demonstrated and the findings of Liu et al. should be stated here in numbers."*

**Response:** Accepted. We add the detailed result given by Liu et al. (2018). Please see Line 171-172.

*Referee: "Line 138: Again, you need to explicitly give the findings of Li et al. that actually agree with your 10microgram m-3 finding."*

**Response:** Accepted. We show the findings of Li et al (2017) to support our results. Please see Line 166-169.

*Referee: "Line 141: What percentage of the wind is downwind per annum?"*

**Response:** Accepted. The long-range transport of aerosols in China is dominated by the westerly winds, which are prevailing winds from the west toward the east in the middle latitudes between 30 N and 60 N (Liang et al., 2004). Because our simulations were conducted on a global scale, the quantitative impacts of China's emission variations on the aerosol distributions in different downwind regions can be seen clearly in Fig. 2. We improve the sentence. Please see Line 175-176.

*Referee: "Line 142: verity->verify."*

**Response:** Accepted. We modify this word. Please see Line 177.

*Referee: "Line 146: You cannot find a trend by two years only. If you had all the years between 2008 and 2016, i.e. 9 years, then you might be able to claim this. Your simulations show a difference between two years, this is not a trend."*

**Response:** Accepted. The results mentioned here are derived from the simulations in 2008 and 2016. Nevertheless, the variations of BC and sulfate in the model are in line with exiting observations, which have demonstrated the decreasing trends over the past decade (Liu et al., 2018; Kanaya et al., 2020). We reword the sentence. Please see Line 177-183.

*Referee: "Line 147: You need to mark this island for those of us who do not know where it is and show in a figure this decline. Again, if you are showing results from two years only this is not an "interannual" decline, this is a decline between two years."*

**Response:** Accepted. The decline of BC concentrations in Fukue island between 2008 and 2016 has been noted in Figure 1c, d. The long-term observations in Fukue island show a significantly decreasing trend in BC concentrations (Kanaya et al., 2020) and support our simulation results.

*Referee: "Line 150: This fact was not made clear in a satisfactory manner to the audience. You need more precise statistics."*

**Response:** Accepted. As suggested by the referee, we add more detailed statistical data for the comparison between simulations and observations as well as previous studies. We also reword this sentence. Please see Table S3, Line 162-190.

*Referee: "Figure 1: You mean, the locations of the EANET stations are shown, not the actual numbers these provide. Replace literatures with literature."*

**Response:** Accepted. We replace the word literature used here. The observed aerosol concentrations (the values) at EANET stations are shown in Figure 1. The inner colors of each circle represent the magnitude of aerosol concentrations, which are overlaid on the map of simulation results. The detailed location information of EANET stations is given in Table S1.

*Referee: "Figure 2: This is not a map of China, this is a global map. Does this mean you run your simulations for the entire globe? this should be made clear in Section 2.1."*

**Response:** Accepted. We run the global climate model on a global scale. Please see Line 103-104.

*Referee: "Line 162: Not shown clearly. Impossible to tell what colour the dots have."*

**Response:** Accepted. The comparison of simulated nitrate concentrations with observations can be seen in both Table S2 and Figure S3. The observed concentrations are shown in colored dots, which are overlaid on the map of simulations.

*Referee: "Line 165: Similar, i.e.? in numbers?"*

**Response:** Accepted. The numbers are added here. Please see Line 196-198 in the revised manuscript.

*Referee: "Line 182: Not a trend if only two years are used."*

**Response:** Accepted. We reword the phrase in this section. Please see Line 211.

*Referee: "Line 184: This is a very important finding and should be included in the main paper."*

**Response:** Thanks for the suggestion. Because similar findings for the comparison between satellite-observed and modeled AOD have been reported in several previous studies (He et al., 2015; Sockol and Small Griswold, 2017), we only display those results in the Supplementary Materials (Figure S4 and S5).

*Referee: "Line 187: This can be due to a million different reasons. You do not give an std, maybe the respective errors compensate for this difference. Also, how exactly did you calculate the annual mean AOD from the two sets of information? did you collocate on a daily basis, then created the monthly and then the annual? another way? these simple choices may affect your numbers greatly."*

**Response:** Accepted. We show the standard deviations (i.e., uncertainties ranges given by the MISR product, please see Garay et al., 2020) and inter-annual trends in AOD values from MISR. The MISR aerosol product at a one-year temporal resolution is used in this study. Please see Figure S6 and Line 155-159 and Line 227-228 in the revised manuscript.

[Figure]

Fig. S6 Inter-annual variation in AOD at 550 nm retrieved by MISR over East China during 2008–2016. The mean values (dots) and standard deviations (vertical lines) are calculated using the gridded AOD and standard deviation from the Level-3 MISR product with a one-year temporal resolution. The black line represents the linear fit of the AOD (T-test: α=0.013<0.05).

*Referee: "Line 188: How negative? precisely."*

**Response:** Accepted. The numbers are added in the statement. Please see Line 218-220.

*Referee: "Figure 4: So, these are the global changes due to the local emission variation? which experiment is this, from Table1? the first?"*

**Response:** Accepted. The changes are derived from the Exp08 and Exp16 simulations. All the experiments are performed on a global scale but only results in China and outflow areas used for analysis. Figure 4 shows the changes in East Asia and continental outflows. Please see Line 103-115.

*Referee: "Figure 4: Where are the coloured dots?"*

**Response:** Accepted. We update Figure 4 to show the location of the AERONET station. The detailed values of observations and the comparison with simulations have been given in the main text. Please see Line 237-242.

*Referee: "Line 206: That may be the case, but from the material you provide, it cannot be verified."*

**Response:** Accepted. More details on the model evaluation on aerosol optical properties and associated uncertainties are added in the section. We reword this sentence here based on the evidence we provided. Please see Line 222-246.

*Referee: "Line 244: You need to show these results."*

**Response:** Accepted. The results have been shown in this sentence. Please see Line 285-287.

*Referee: "Figure 5: Why not zoom these maps to show Asia only? more detail would appear and it is clear that the rest of the world is not affected."*

**Response:** Accepted. We zoom out the Figure to underline the variations in Asia and surrounding areas. Please see Figure 5 and Figure 6 in the revised manuscript.

*Referee: "Figure 5f: This map is impossibly noisy, neighboring pixels show -1 and 1 Wm-2 forcing, this is not physical."*

**Response:** We would like to point out that the response of cloud radiative forcing to aerosol perturbations is distinguishable from the noise over China and north Pacific (Figure 5f). The aerosol-induced cloud radiative forcing (CRF) is estimated by varying the anthropogenic emissions between 2008 and 2016 in two different simulations. The CRF noise can be caused by the aerosol-meteorology feedback in the global climate model. Such anomalies of aerosol-induced CRF signals are also seen in similar studies (e.g., Shi and Liu, 2019).

**Abstract.** Anthropogenic emissions in China play an important role in altering global radiation budget. Over the recent decade, the clean-air options in China resulted in substantial reductions in anthropogenic emissions, such as sulfur dioxide ($SO_2$) and primary carbonaceous aerosols, and consequently improved air quality. However, the resultantassociated changes aerosol radiative forcingsin aerosol climate effects are poorly understood and few reported. In this study, we use an advanced global
10 climate model integrated with the latest anthropogenic emission inventory to estimate the changes in the aerosol radiative forcings by the anthropogenic emission variation in China between 2008 and 2016. OFirst, our simulations exhibit decreases of 46% and 25% for the annual mean surface-level sulfate and black carbon (BC) mass concentrations in East China, respectively, which is the key region subject to stringent emission control options. In the meantime, Tthe simulated aerosol optical depth and aerosol absorption optical depth show decreasing tendencies. Results reveal that Tthe substantial reductions
15 in $SO_2$ emissions yield a positive all-sky shortwave aerosol direct radiative forcing of +0.17 W m$^{-2}$ at the top of the atmosphere (TOA) primarily through the weakening of sulfate scattering effects, and an aerosol-induced cloud radiative forcing of +0.13 W m$^{-2}$ in East China during 2008–2016. The reduction in BC emissions induces a negative BC radiative forcing of −0.34 W m$^{-2}$. Hence, that the positive radiative forcing by the $SO_2$ emission reductions may beis counterbalanced by those of the decrease of BC concentrations in China during 2008–2016. Besides, the model experiments show a clear enhancement of the
20 downward solar radiation flux that reach the Earth's surface over China caused by the mitigation of aerosol pollution, agreeing with long-term observation records of the shortwave energy balance from recent studies. While the radiative forcing at TOA is small locally due to the counteracted effects of $SO_2$ and BC emissions, it is relatively larger (+0.16 W m$^{-2}$) over the north Pacific remote regions for this period, primarily contributed by the reductions in sulfate particles and their effects on cloud properties. By adoptingWith a comprehensive future emission scenario for China in 2030 and 2050 developed by the recent
25 study, we predict that the strictest environmental policies will induce the change of aerosol radiative forcings of +0.55 and +1.23 W m$^{-2}$ over East China between 2016−2030 and 2016−2050, respectively. Since aerosol radiative forcings potentially influence surface temperature, boundary layer dynamics, and precipitation through fast climate responses, tailoredTargeted emission control policies are desirable to improve air quality and mitigate the risk of climate change in the future.

**1. Introduction**

Aerosols perturb the global energy balance by aerosol-radiation interactions, such as the scattering and absorption of sunlight (Charlson et al., 1992), and by aerosol-cloud interactions through the activation of cloud condensation nuclei (CCN) particles into cloud droplets, which impact on both the cloud albedo and lifetime (Twomey, 1974; Andreae and Rosenfeld, 2008). The changes in anthropogenic aerosol concentrations from preindustrial to present days are estimated to induce a global-mean net cooling effect of −0.4 to −1.5 W m$^{-2}$ at the top of the atmosphere (TOA) that partly mask the warming effects by increased carbon dioxide (Boucher et al., 2013).

It is commonly known that black carbon (BC) and sulfate aerosols are important contributors to the radiation absorption and scattering effects of anthropogenic aerosols in the global scale. Bond et al. (2013) estimated that the industrial-era (1750−2005) direct radiative forcing of BC is 0.71 W m$^{-2}$. Until now, uncertainties embedded in the radiative forcing of BC are still large due to the insufficient treatment of BC atmospheric processes in climate models including the impacts of BC on liquid clouds (Koch and Del Genio, 2010; Chung and Seinfeld, 2002) and the role of BC in acting as ice nuclei (Kulkarni et al., 2016). Moreover, the mixing state is one of the key parameters that determine the optical properties and CCN activity of BC (Jacobson, 2001; Stier et al., 2006; Matsui, 2016). Recent studies find that explicit representation of BC aging processes can increase the confidence in the estimates of BC direct radiation forcing (Matsui et al., 2018a). Unlike BC that is directly emitted into the atmosphere, sulfate aerosols mainly originate from the chemical transform of sulfur dioxide (SO$_2$) via the photochemical oxidation by OH radical, aqueous and heterogeneous reactions (Seinfeld and Pandis, 2016). The accurate estimate of sulfate radiative forcing relies heavily on the representation of secondary sulfate formation in climate models and SO$_2$ emissions (Huang et al., 2015). Sulfate aerosols are estimated to exert a global-mean direct radiative forcing of −0.32 W m$^{-2}$ for the time period of 1750 to 2010 (Myhre et al., 2013), with remarkable radiative perturbation in the north mid-latitude region (20 °−40 ° N) due to the rapidly increased anthropogenic SO$_2$ emissions in China over the past few decades. The tremendous anthropogenic emissions in China not only result in severe air pollution, but also significantly alter the global aerosol radiation budget (Li et al., 2016a).

During the past ten years, China has implemented stringent air pollution control measures and the SO$_2$ emissions started to decrease in 2007 with the application of flue gas desulfurization in the power sector. Especially since 2013, the toughest-ever clean air policies have led to substantial reductions in anthropogenic emissions in China. According to the latest emission inventory, the national annual emissions of SO$_2$, NO$_x$, BC, and organic carbon (OC) have declined by 62%, 17%, 27%, and 35%, respectively during 2010−2017 (Zheng et al., 2018). Recent studies have demonstrated significant improvements of air quality in China attributable to those various emission control measures (Zhang et al., 2019). Specifically, SO$_2$ emissions exhibited the most notable reduction among all pollutants for this period, which reduce the concentrations of sulfate aerosols dramatically and mitigate the PM$_{2.5}$ pollution and acid rain issues (Liu et al., 2020; Liu et al., 2018).

The unprecedented reductions in anthropogenic emissions over China provide a unique opportunity to investigate the responses of aerosol climate effects to such high variations in their precursor emissions. Those emissions changes would lead to the

perturbation of the Earth's solar radiation budget, consisting of the aerosol forcings at the top of atmosphere due to aerosol-radiation interactions and aerosol-cloud interactions. Fadnavis et al. (2019) estimate that the Chinese $SO_2$ emissions reduction during 2006–2017 produces a positive clear-sky direct radiative forcing of +0.6 to +6 W m$^{-2}$ over China. Paulot et al. (2018) have investigated the trends in the aerosol radiative effects in eastern China from 2001 to 2015 and shown a clear decreasing tend in AOD starting from 2007 due to reductions in $SO_2$ emissions. Previous estimates of aerosol forcings may be inadequate since they adopt athe clear-sky condition and simple treatment of the mixing between sulfate and BC and do not consider the aerosol-cloud interaction in their simulations, all of which are important in the calculation of aerosol total radiative effects (Ghan, 2013). Until now, it remains unclear how the overall changes in anthropogenic emissions in China in the past decade (from 2007 to the present) including not only $SO_2$ emissions but also BC and other aerosol components impact aerosol forcings in source regions and outflows. BC emissions are reported to be significantly reduced in eastern China from 2010 to 2019 as inferred from rapidly decreased BC concentrations observed at an in-situ site near China mainland (Kanaya et al., 2020). Because BC particles exert a positive radiative effect through direct absorption of solar radiation and absorption enhancement by non-BC particles (like sulfate and organics) in the atmosphere(Matsui, 2020), the radiative effects caused by the changes in BC and non-BC emissions should be quantified in detail. Estimates of aerosol radiative forcings can favor a better prediction of the climate responses of future emission scenarios, like the shared-economic pathways (SSPs), on a regional or global scale.

In this study, we aim to evaluate the response of aerosol radiative forcings (RFs) including both direct radiative effects and aerosol effects on clouds to the change in anthropogenic emissions in China between 2008 and 2016. We chose these two years for analysis because the differences of emissions between these two years (2008 and 2016) can reflect the inter-annual changes in China's anthropogenic emissions caused by a series of national emission control measures over the past decade (Li et al., 2017; Zhang et al., 2019). Model experiments using Aan advanced global climate model integrated with the latest bottom-up emission inventory for China are performed is used to diagnose the changes in RFs from different aerosol components (sulfate, nitrate, BC etc.) and their interactions with radiation and clouds. Furthermore, wWe perform another experiment tofurther predict the radiative effects of aerosols with projected emission scenarios for the years 2030 and 2050.

**2. Methods**

**2.1 Model experiments**

In this study, we use the Community Atmospherete Model version 5 (CAM5) with the Aerosol Two-dimensional bin module for foRmation and Aging Simulation version 2 (CAM5/ATRAS2) (Matsui and Mahowald, 2017; Matsui, 2017). The release version of CAM5 can simulateion emissions, gas-phase chemistry (MOZART), aerosol microphysical and secondary formation processes, wet/dry deposition, and aerosol-radiation-cloud interactions (Liu et al., 2012). The ATRAS2 module (Matsui et al., 2014) uses a two-dimensional sectional representation with 12 particle size bins (from 1 to 10000 nm in diameter) and 8 BC mixing state bins (from fresh BC to aged BC-containing particles) for various microphysical and chemical processes

of aerosols, including new particle formation, condensation/coagulation, aerosol activation, wet/dry deposition, and interactions with radiation and clouds. The model treats aerosol–cloud interactions in stratiform clouds using a physically based two-moment parameterization that considers the aerosol effects on cloud properties (Morrison and Gettelman, 2008). The secondary formation of sulfate is found to be important when simulation sulfate concentrations in China, but is not represented well in current chemistry-climate models (Hung and Hoffmann, 2015; Cheng et al., 2016). To better reproduce the temporal evolution of sulfate concentrations in China, we add a new pathway suggested by previous studies for secondary sulfate formation in our model, that is, the heterogeneous oxidation of gaseous $SO_2$ to particulate sulfate onto aerosol surfaces under high humidity conditions (Huang et al., 2014). The uptake coefficients of $SO_2$ are specified as the range between $2.0 \times 10^{-5}$ and $1.0 \times 10^{-4}$ under the ambient relative humidity of 50−100%.

The model is running at the horizontal resolution of $1.9\,° \times 2.5\,°$ with 30 vertical layers from the surface to ~40 km on the global scale. Several simulation experiments are designed with different inputs of anthropogenic emissions as shown in Table 1. We vary the anthropogenic emissions of all species for years  2008 and 2016 in China, termed as Exp08 and Exp16, respectively. The Exp16SO2 and Exp16BC cases are made by replacing the $SO_2$ and BC emissions in the Exp08 with those for 2016, respectively. Note that the radiative effects by each aerosol component are calculated online by our model and we use the results in the Exp08-Exp16 cases for discussing the main conclusion and implications. The cases of Exp16SO2 and Exp16BC with only one of the emission change included are used for comparison with the Exp16 to reflect the interactions of BC and $SO_2$ emission variations. We perform 2-year simulations for each experiment and the first year is for spin-up and the second year is for analysis. The meteorological fields were nudged by using the Modern-Era Retrospective analysis for Research and Applications Version 2 (MERRA2) data and were fixed at the year 2008 in these experiments to  separate the contribution of emissions on aerosol radiation effects from variation in meteorological fields. The inter-annual change in meteorological conditions could influence aerosol radiative forcings but is not the focus of this study, and we simply discuss their importance in Section 3.3.

We diagnose both the  direct radiative effects of aerosols (DRE) and cloud radiative effects (CRE) in the model. The changes in DRE and CRE between the emissions years  2008 and 2016 can be regarded as aerosol  (direct radiative forcing (DRF) and aerosol-induced cloud radiative forcing (CRF)) owing to the variation of anthropogenic emissions . Specifically, the aerosol DRE at TOA for each specific aerosol component (e.g., sulfate, BC and others) is online calculated as the differences between the standard radiative fluxes and the diagnosed one  that subtracting this species in the radiation module. Then the aerosol DRF is calculated as the difference of DRE between 2008 and 2016. The aerosol effects on clouds are diagnosed following Ghan (2013) using the variables of clean-sky (neglecting the scattering and absorption of all aerosol species) radiative flux. The CRF values are scaled to the best estimate ($-0.5$ W m$^{-2}$) of effective radiative forcing due to aerosol-cloud interactions given by the Fifth Assessment Report of the Intergovernmental

Panel on Climate Change (IPCC AR5) (Boucher et al., 2013). This process yields a reasonable estimate of CRF based on our model results and present knowledge of aerosol-cloud interactions from the latest IPCC report.

**2.2 Anthropogenic emissions for China**

The global anthropogenic emissions for the year of 2008 are taken from Hoesly et al. (2018) based on the Community Emissions Data System (CEDS), while the emissions in China are replaced by the Multi-resolution Emission Inventory (MEIC) for 2008 or 2016, which provide a more realistic representation of China's emissions from fossil fuels and biofuels (Zheng et al., 2018). The global emissions other than China used in all simulations are taken from the CEDS for 2008. In addition, the ammonia emissions in China are taken from Liu et al. (2018), in which the estimates of agricultural ammonia emissions are well constrained and show good performance in the simulation of atmospheric ammonia. As shown in Table 2, several major species including BC, primary organic carbons aerosols (OC), and $SO_2$ experience reductions of emissions between 2008 and 2016 due to the clean-air policy in China in this period. The majority (more than 90%) of those emission reductions are found in East China (marked in Fig. 1), which is characterized with dense population and economic activities and is the key region subject to stringent air pollution control. For instance, emission reductions in East China are 57% for $SO_2$, 27% for BC, and 30% for OC.

**2.3 Observation data**

In order to evaluate the model performance for aerosol simulations, we collect the monthly measurement concentrations of inorganic aerosol chemical components (sulfate, nitrate, and ammonium) from Acid Deposition Monitoring Network in East Asia (EANET). The database has been widely used for the studies of air pollution and acid deposition for the region (Itahashi et al., 2018). The EANET stations used in this study include ing one site in southern China and nine sites in western and central Japan. We also obtain the observation data of aerosol compounds for the period 2008–2016 from different measurement campaigns and the yearly averages of them are used to indicate the decadal variation of major aerosol components over China (Table S2). For comparison, the annual mean simulation results are taken from the model surface-level horizontal grid closest to these observation sites. In addition, we use observations of vertical profiles of BC concentrations from the HIAPER Plow-to-Plow Observations (HIPPO) aircraft campaigns conducted in the north Pacific region (Wofsy, 2011). The HIPPO data are obtained from five deployments across the central Pacific during 2009−2011 and include measurements of BC mass concentrations from a single-particle soot photometer (SP2) instrument (Schwarz et al., 2013). The monthly mean BC concentrations in the simulation for 2008 are used for comparison with observations from the same month in five HIPPO campaigns despite the differences in the years selected between the model and observations (Fig. S1).

We employ measurements of aerosol optical depth (AOD) and single scatter albedo (SSA) from use Aerosol Robotic Network AERONET (AERONET Aerosol Robotic Network) surface stations to reflect variations of aerosol optical properties during 2008–2016. The available stations that have long-term data records are Beijing (39.98 °N, 116.38 °E) and Xianghe (39.75 °N,

116.96 °E). In addition, the AOD at 550 nm retrieved by the Multi-angle Imaging Spectroradiometer (MISR) Level-3 product (Garay et al., 2020) and the dark target and deep blue combined 550 AOD by the Moderate Resolution Imaging Spectroradiometer (MODIS) aboard NASA's Terra (Sayer et al., 2014) are used for comparison with our model results. Both satellite products provide the annual-mean or monthly-mean AOD data at a horizontal resolution of 0.5 °×0.5 °that can reflect the temporal changes over China during 2008–2016.

**3. Results**

**3.1 Changes in the burdens of sulfate and BC mass**

We first evaluate the simulated sulfate mass concentrations with the observations in China and Japan between the emission years of 2008 and 2016 (Fig. 1a, b). The simulated sulfate using CAM5/ATRAS2 model generally agrees well with available observations with respect to the magnitude and spatial patterns, showing high annual mean concentrations up to 20 μg m$^{-3}$ in East China in 2008 and decreasing to less than 10 μg m$^{-3}$ in 2016 due to the reductions of about 60% in SO$_2$ emissions (Table S2). Greater reductions of more than 10 μg m$^{-3}$ are found in northern part of East China (Figure 1a, b), where the SO$_2$ emission reductions are the most notable in the country (Li et al., 2017). The observed SO$_2$ loadings given by Li et al. (2017) show severe SO$_2$ pollutions (up to 2.0 DU, 1 DU = 2.69 ×10$^{16}$ molecules cm$^{-2}$) before 2010 in the northern China and significant mitigation in 2016 with the column as low as about 0.5 DU. Despite the lack of sulfate monthly observations, the model satisfactorily captures the seasonal patterns of SO$_2$ amounts retrieved by the Aura Ozone Monitoring Instrument (Fig. S2). The variation of annual mean sulfate concentrations in East China is on average of −3.5 μg m$^{-3}$ (−46%) from 2008 to 2016, which is close to another model result (−50%) within the region (Liu et al., 2018). In the meantime, The sulfate mass column burden (integral of concentrations from surface to the top of atmosphere) averaged over East China exhibits a decrease of 4.5 mg m$^{-2}$, equal to 35% of that burden in 2008 (Fig. 2a, c). Moreover, the averaged sulfate burden in the northern Pacific (the region marked in Fig. 2) decreases on average by 0.38 mg m$^{-2}$ (−18%) in the northern Pacific (the region marked in Fig. 2), reflecting the impacts of SO$_2$ emission reductions from East China to the downwind regions dominated by the westerly wind field in the middle latitudes.

We verifty the trend variation of in modeled BC concentrations using the measurements in three typical sites located in East China and western Japan (Fig. 1c, d). The long-term observational records of BC concentrations in Beijing and Shanghai present clear decreasing trends over the recent decade (Xia et al., 2020; Wei et al., 2020). For the two time slice (i.e., 2008 and 2016), Tthe annual mean surface BC decreases from 8.5 μg m$^{-3}$ in 2008 to 3.5 μg m$^{-3}$ in 2016 in Beijing, and from about 4.0 μg m$^{-3}$ to 2.2 μg m$^{-3}$ in Shanghai. Similar reductions are shown in our simulations for the emission years of 2008 and 2016 (Fig. 1c, d), with the averaged decrease of 25% over East China. The continuous decline (−5.8±1.5) % yr$^{-1}$) in surface BC concentrations observed at Fukue Island in western Japan from 2010 to 2018 further supports the reduction of the BC emissions in East China (Kanaya et al., 2020). Moreover, we evaluate the vertical distributions of BC concentrations in the northern

Pacific using the BC measurement by HIPPO campaigns (Fig. S1). Despite some underestimation of modeled BC in a few months, there is in general a reasonable agreement between modeled and observed BC concentrations from surface to the upper troposphere. During 2008–2016, the annual mean BC burdens decrease by 0.38 mg m$^{-2}$ (22%) for East China and by 0.016 mg m$^{-2}$ (13%) in the northern Pacific, respectively (Fig. 2b, d). . In general, the results suggest that the integration of CAM5/ATRAS2 model and the MEIC emission inventory for China can reproduce the observed decline in the sulfate and BC concentrations over the recent decade caused by the variations of anthropogenic emissions.

 The simulated particulate nitrate concentrations are compared well with the measurements (Fig. S3 and Table S2). We find that the annual mean nitrate concentrations are elevated by about 1.0 μg m$^{-3}$ (20%) in East China between emission years 2008 (Exp08) and 2016 (Exp16). The nearly consistent variation takes place in the Exp16SO2 case. Since NH$_3$ emissions are unchanged, the variation in SO$_2$ emissions is responsible for the increases of particulate nitrate concentrations. The reductions of ammonium sulfate particles release free ammonia in the air to facilitate the partitioning of  nitrate toward the aerosol phase (Ansari and Pandis, 1998). Similar increase (1–2 μg m$^{-3}$, 28%) of surface nitrate concentrations within the region has been reported by Liu et al. (2018) for the period using a regional chemical transport model. Such a relationship between SO$_2$ emissions and nitrate concentrations in China has been also pointed out by Leung et al. (2020). In addition, the annual mean concentrations of particulate ammonium and organic aerosols in East China are reduced by 0.41 μg m$^{-3}$ and 0.13 μg m$^{-3}$, respectively.

The simulations also reveal that the reductions in sulfate and BC burdens would dampen new particle formation and the growth of particles via condensation and coagulation processes to serve as CCN. Figure 3 shows the longitude-height (pressure in hPa) distribution of the changes (%) in particle number concentrations with the diameters larger than 10 nm (N10) and diagnostic CCN numbers at supersaturation of 0.4% (CCN0.4) along the latitude 35 °N, which is characterized with notable reductions in sulfate and BC burdens. The N10 shows decreases of about 20% in East China (100−130 °E) from the surface to 300 hPa. The simulated CCN0.4 concentrations decrease not only in East China (30%) but also in the middle troposphere (700−200 hPa) over the north Pacific region (10%). According to the sensitivity experiment Exp16SO2, these variations in N10 and CCN0.4 are primarily accounted by decreases of particulate sulfate in East China and the downwind regions. The decreases of CCN concentrations would alter the cloud properties through the activation into cloud droplets and subsequently change the radiation budget, which is discussed in the next sections.

**3.2 Variations in aerosol optical properties**

The pronounced variations of sulfate and BC concentrations between 2008 and 2016 would induce the change in aerosol optical properties, i.e., aerosol optical depth (AOD) and absorption aerosol optical depth (AAOD) at 550 nm. The CAM/ATRAS2 model captures the observed hotspots of AOD (> 0.3) over North Africa, East China, the tropical Atlantic Ocean, and West-South Asia by comparing to the MISR and MODIS measurements (Fig. S4 and Fig. S5). The annual

mean AOD for East China in the model (0.25) is lower than that from MISR (0.34) and MODIS (0.41) in the year of 2008. The comparison bias global of regional AOD distributions with satellite-observed measurements are similar to previous CAM model modeling studies by CAM models (Sockol and Small Griswold, 2017; He et al., 2015). For instance, Sockol and Small Griswold (2017) present the difference between averaged AOD predicted by the CAM models and MODIS of < −0.2 over East China. The model biases found in some continental outflow areas in the tropics and South Hemisphere may be related to inadequate simulations of sea salt aerosols and their optical properties in the marine atmosphere (Bian et al., 2019; Burgos et al., 2020).

Figure 4 presents the differences of simulated AOD and AAOD these optical variables at 550 nm between 2008 and 2016. Corresponding to the reductions in sulfate and BC mass concentrations, AOD shows percent decreases of 2−10% over most of East China (marked in black box in Fig. 4), with a region average of on average of 5.2%., in which tThe variation in AOD from induced by sulfate particles decreases by 35% and has a dominant contribution to the totalAOD variation. Recent studies find a statistically significant decrease of 10−20% for AOD in the same region over the recent decade based on the remote sensing observations from the surface (AERONET) and space (MODIS and MISR (Fig. S4)) (Li, 2020; Zhang et al., 2017; Zhao et al., 2017). The annual mean AOD retrieved by MISR varied from 0.34 ±0.07 in 2008 to 0.26 ±0.06 in 2016 and shows a significantly decreasing trend during the period (Fig. S6), while the simulated AOD in the CAM5/ATRAS2 model decreases by about 0.02 between the two years. The underestimation of AOD trendsvariability may be partially attributable related to the negative bias of modeled AOD as abovementionedthe inadequate representation of decadal variations in dust aerosol concentrations from both natural and anthropogenic sources. Recent studies have found a considerable decline in dust episodes over East China during 2007−2014 due to decreased maximum wind speeds and control of air pollution (Wang et al., 2018), and the resulting decrease of dust loadings and their light extinction in air would contribute to the decreases of total AOD, but those effects cannot be fully represented in our simulations. Moreover, the parametrization of aerosol hygroscopicity, especially for light-scattering components (sulfate, nitrate, OA, etc.), can highly determine modeled aerosol extinction coefficients and thus is an important source of uncertainties in the estimate of AOD in climate models (Burgos et al., 2020; Reddington et al., 2019).

The simulated annual mean AAOD decreases by about 16% (from 0.018 to 0.015) over East China between the emission years of 2008 and 2016 (Fig. 4), with salient decrease up to 40% in the northern part, which is subject to high reductions of BC emissions and resultant column burdens (Fig. 2). We compare the model results with the averaged observations in two AERONET stations (located in the same model horizontal grid and labelled marked in Fig. 4) with long-term records available. The AAOD from AERONET are calculated using the observed AOD and SSA at 550 nm wavelength. The annual mean observed AOD and AAOD at the AERONET site decrease by about 29% and 40%, respectively, while our model underestimates the variability of AOD, but agrees well with observed AAOD variation. Overall, both the simulations and observations demonstrate a clear decline in AOD and AAOD over China between 2008 and 2016 and the reductions in sulfate

250 and BC particles play a critical role in the phenomenon. The resulting effects on the solar radiation budget over China and outflow areas are discussed in the next section.

**3.3 Radiative forcings by the change in anthropogenic emissions between 2008 and 2016**

Here, we focus on the all-sky aerosol RFs at TOA by the inter-annual variation in anthropogenic emissions over China between 2008 and 2016, especially for East China. Both the aerosol shortwave DRF (also termed as RFs due to aerosol-radiation

255 interactions) and shortwave + longwave aerosol effects on CRF (including the semi-direct and indirect effects in this study) are considered. As shown in Fig. 5 and Fig. S7, the decrease of sulfate mass burdens due to the substantial reduction in $SO_2$ emissions induce a positive radiative forcing by diminishing the scattering effects of sulfate aerosols. The mean sulfate DRF is +0.38 W m$^{-2}$ over East China during the study period, with high values of more than 0.6 W m$^{-2}$ in the north part that experiences the most notable reduction in sulfate burdens. As a result of $SO_2$ emission reductions, nitrate concentrations have

260 increased, and the merged nitrate and ammonium DRF is estimated to be −0.21 W m$^{-2}$, though the decrease of particulate ammonium concentrations should yield a positive RF.

The reduction in BC emissions would diminish the absorption effect of BC particles with the decline of the annual mean BC mass burden of 22% over East China between 2008 and 2016 (Fig. 5b and Fig. S8). The resultant BC DRF in the Exp16 case is −0.34 W m$^{-2}$ in the region, accounting for 19% of the BC DRE in 2008. This is 17% higher than the estimate of BC DRF

265 derived from the simulation scenario with BC emission change only (Exp16BC), That is because the concurrent declines in sulfate concentrations and associated water uptake weaken the absorption enhancement of BC-containing particles, and thus further diminish the positive radiative effects of BC. The absorption enhancements by BC coating materials will be overlooked if BC is externally mixed with other aerosols in climate models, which will significantly underestimate the magnitude of the BC DRF for China between 2008 and 2016. Moreover, the treatment of aging processes of BC is another important factor in

270 determining the radiation effects of BC-containing particles. The models with single mixing state for BC-containing particles (without considering the aging processes from fresh BC to thickly-coated BC) could overestimate (underestimate) the BC DRF when BC-containing particles are assumed to be internally mixed (externally mixed) with other components immediately after emissions (Bond et al., 2013; Matsui et al., 2018a).

We find that the total aerosol DRF due to all components is −0.18 W m$^{-2}$ over East China between the emission years of 2008

275 and 2016, with the dominant sources from the reductions in $SO_2$ and BC emissions (Fig. 6a). Noticeably, the BC and sulfate DRF (−0.34 vs. +0.38 W m$^{-2}$) are almost counterbalanced by each other. After considering the negative nitrate DRF (−0.21 W m$^{-2}$) resulting from the $SO_2$ emission reduction, the net DRF from $SO_2$ emissions between 2008 and 2016 is +0.17 W m$^{-2}$. OA and dust aerosols have the mean DRF of −0.06 and +0.003 W m$^{-2}$ and contribute few to the total aerosol DRF during the study period. A few uncertainties remain in our estimates of aerosol DRF. The model bias in the AOD variability compared to the

280 satellite observations could transform into the uncertainties in aerosol DRF. As mentioned early, the model underestimation of AOD variability during the study period might stem from the uncertainties in dust emissions (natural and anthropogenic

sources) and aerosol water uptake by hygroscopic components that are not well represented in current climate models (Burgos et al., 2020). Because the AAOD variation is simulated reasonably, the aerosol scattering effects contributing to AOD from dust, sulfate and OA may be underestimated in our simulations.

We further consider aerosol effects on CRF with the combination of semi- and in-direct effects, following Ghan (2013). The variation of BC and $SO_2$ emissions have distinct impacts on cloud radiative effects. First, as mentioned in Section 3.1, the responses of CCN numbers are negligible to the reduction in BC emissions, but of significant to the reduction in $SO_2$ emissions with a decrease of 20% for the CCN number concentrations at 0.4% supersaturation. The changes in $SO_2$ emissions and associated decline in sulfate would alter in-direct effects by serving as CCN due to its high hygroscopicity (Twomey, 1974; Andreae and Rosenfeld, 2008). By separating the contribution between $SO_2$ and BC emissions to CRF (the Exp16SO2 and Exp16BC cases), we find that the annual mean CRF is +0.16 W m$^{-2}$ due to the decreased $SO_2$ emissions in East China between 2008 and 2016, whilst only −0.05 W m$^{-2}$ due to decreased BC emissions. The positive CRF can be explained by the decrease of simulated CCN numbers (Fig. 3), and it is indicative of less solar radiation outgoing back to the space by clouds and therefore potentially warming the atmosphere. By accounting for the interaction between sulfate and BC particles, we find a net aerosol-induced CRF of +0.13 W m$^{-2}$, and the total RF of that combines aerosol DRF and CRF together is −0.04 W m$^{-2}$ over East China during our study period (Exp08 - Exp16).

While the impacts of $SO_2$ and BC reductions on RFs (DRF + CRF) are very small locally (−0.04 W m$^{-2}$), there are markedly positive cloud radiative effects mostly within 0.3−0.6 W m$^{-2}$ in the north Pacific region (Fig. 5), with the mean CRF over the north Pacific (marked in Fig. 5f) of +0.16 W m$^{-2}$, which are associated with the decrease of CCN due to reductions in $SO_2$ emissions in China (Figs. S7). Since the aerosol DRF in the north Pacific region is negligible, the aerosol effects on cloud dominate the total radiative effects when the anthropogenic emissions in China varied from 2008 to 2016. Our results suggest an important role of $SO_2$ emissions in China in altering the cloud optical properties in the north Pacific. It should be noted that aerosol effects on cloud radiative properties remain the largest uncertainties in calculating global and regional radiation perturbation by anthropogenic aerosols in current climate models, which should be further constrained by measurements of cloud properties.

Furthermore, the inter-annual variations in scattering and absorbing aerosols during 2008–2016 lead to changes in downward radiation flux onto the Earth's surface (Fig. 6). Evident enhancement in surface radiation flux (i.e., surface brightening) take place in East China. The decreases of BC absorption and sulfate scattering effects allow more solar radiation reaching the surface with the temporal changes estimated of +0.96 W m$^{-2}$ decade$^{-1}$ and +0.75 W m$^{-2}$ decade$^{-1}$, respectively (Fig. 6b, c). Increases of nitrate and OA concentrations diminish the downward surface solar shortwave radiation. The net change in the annul-mean surface radiation due to all aerosols is as large as 1.1 W m$^{-2}$ decade$^{-1}$ (Fig. 6a). Compared to those induced by aerosols-radiation interactions, the aerosol-induced cloud effects on surface radiation are minor (+0.20 W m$^{-2}$ decade$^{-1}$). With the consideration of the overall effects by the differences in emissions between 2008 and 2016, the simulated downward

radiation fluxes onto surface increase by +1.3 W m$^{-2}$ decade$^{-1}$. Our results agree with the long-term observational trend of the shortwave energy balance in China from 2000–2015 provided by Schwarz et al. (2020). They have shown a positive trend in downward surface shortwave radiation of +1.4 ±0.2 W m$^{-2}$ decade$^{-1}$ for the period, which is considered to be mainly driven by the reductions in the atmospheric shortwave absorption. The decadal change in solar absorption of BC is estimated to be −1.4 W m$^{-2}$ decade$^{-1}$ in our simulations, 56% lower than the observed total atmospheric shortwave absorption (−3.2 ±0.4 W m$^{-2}$ decade$^{-1}$) based on observational data for the similar period (Schwarz et al., 2020). Apart from BC, other absorbing aerosol components, like anthropogenic magnetite (Matsui et al., 2018b) and brown carbon from fossil fuels and biomass or biofuel burning sources (Zhang et al., 2020; Yan et al., 2017) may also contribute to decadal variation of atmospheric shortwave absorption in China. In summary, our simulation results provide the first modeling evidence for that the mitigation of aerosol pollution, particularly of BC and sulfate, critically determines the surface brightening observed in China over the recent decade (Yang et al., 2019; Schwarz et al., 2020).

In order to understand how aerosol DRF and CRF respond to future emissions, we refer to the projected anthropogenic emission scenarios in China recently developed by Tong et al. (2020) for 2030 and 2050. For these scenarios, they have integrated the shared socio-economic pathways (SSPs), climate targets of Representative Concentration Pathways (RCPs), and local air pollution control measures to create dynamic projection of China's emissions in the future decades. Here, among their designed scenarios, we choose the SSP1-RCP26-BHE (Best Health Effect), which denotes the best-available environmental policies and therefore the largest reductions of emissions, to predict the upper bound of climate effects from anthropogenic aerosols. We performed the simulations by using the same meteorological nudging with the historical case for 2008 (Exp08) and scaling anthropogenic emissions (BC, SO$_2$, NO$_x$, OC, VOCs, Primary PM, NH$_3$, and CO) in China separately from 2016 to 2030 and from 2016 to 2050 in each grid cell following Tong et al. (2020) (the scaling factors for each species are shown in Table S1S3). The projected emissions of greenhouse gases are not considered and only the aerosol effects on radiation budget are shown here. We find that under the strictest air pollution control policies, the aerosol DRF are +0.24 W m$^{-2}$ between 2016−2030 and +0.64 W m$^{-2}$ between 2016−2050 over East China (Fig. 7). The total radiative effects with the combination of DRF and CRF reach +0.55 W m$^{-2}$ and +1.23 W m$^{-2}$, respectively. For this scenario, although BC emissions keep decreasing in the future, the reductions in scattering aerosols like OA and nitrate induce high positive RFs. The RFs forof OA and nitrate due to aerosol-radiation interactions are +0.27 and +0.52 W m$^{-2}$ in our cases between 2016−2050. Similar to our case, Samset et al. (2019) have estimated a net radiative forcing induced by sulfate and BC aerosols of approximately +0.5 W m$^{-2}$ over China under a strong air-quality policy (SSP1-1.9) from 2014–2030 using the climate model Oslo CTM3 with an oversimplification of aerosol-cloud interaction effects. In addition, the aerosol effects on clouds forcings in north Pacific are around +0.2 W m$^{-2}$ for the both 2016−2030 and 2016−2050 cases and lower than the CRF in East China.

This study mainly focuses on the impacts of emission variations on aerosol radiative forcings, while the meteorological conditions could also affect aerosol forcings. We perform another simulation experiment with the meteorological nudging and anthropogenic emissions both for 2016 (referred to as Exp16m) and compared the resulting aerosol DRF with those found in

the Exp16 case (only emission variation included; see Table S4). Note that in the Exp16m case, the aerosol effects on cloud forcings cannot be separated from the variation in meteorological fields in the model. The changes in BC and sulfate mass burdens for Exp08-Exp16m (differences between the Exp08 and Exp16m cases) are close to those for Exp08-Exp16, but the BC DRF value is reduced by about 29% due to variation in meteorology during 2008–2016. Nevertheless, both our results and previous studies (Liu et al., 2020; Zhang et al., 2019) demonstrate that the decadal variations in aerosol pollutions and DRF are dominated by the reductions in anthropogenic emissions over the past decade.

**4. Summary**

In this study, we quantify the all-sky aerosol RF for China (especially East China) due to the substantial variations of anthropogenic emissions in the country between 2008 and 2016 using a global climate model. Our simulations demonstrate that the dramatic reductions (−57%) in $SO_2$ emissions decrease sulfate mass burdens by 35% in East China and 18% in north Pacific, while the burdens of BC are reduced by 25% due to decreased BC emissions (−27%) for this period. It is estimated that the reductions in $SO_2$ emissions give rise to +0.38 W m$^{-2}$ for sulfate DRF at TOA, −0.21 W m$^{-2}$ for nitrate and ammonium combined DRF, and +0.13 W m$^{-2}$ for aerosol-induced CRF, and meanwhile the BC emission reduction induce a cooling effect of −0.34 W m$^{-2}$ over East China. Since the effects from other aerosol components are negligible in our simulations, the changes in $SO_2$ and BC emissions dominate the aerosol radiative effects forcings between 2008 and 2016. Their net RF by the changes in the BC and SO₂ emissions is −0.04 W m$^{-2}$, implying a counterbalancing effect. It's also interesting to note that the declines in sulfate concentrations and associated water uptake weaken the absorption enhancement of BC-containing particles, and contribute to 17% of the BC DRF compared to that with BC emission changed only. Such an effect cannot be detected without an explicit treatment of absorption enhancement for internally mixed BC particles, as developed in our model (Matsui, 2020).

Moreover, in agreement with the observational-based study on the long-term trend of the downward shortwave radiation that reaches the Earth's surface (Schwarz et al., 2020), our simulations indicate the surface brightening over China from 2008 to 2016. It is estimated that the annul mean surface solar radiation increases by +1.3 W m$^{-2}$ decade$^{-1}$, which is primarily contributed by the diminishment of BC absorption and sulfate scattering effects in the atmosphere. The results provide the direct 
[revised manuscript text omitted]

[Figure]

**Figure 6.** Changes in downward surface shortwave radiation over China and outflow regions between the emissions from 2008 and 2016 due to effects of all aerosols (a), BC (b), sulfate (c), nitrate + ammonium (d), OA + dust (e), and cloud albedo induced by aerosols (f). The positive values indicate the surface brightening and the negative surface dimming.

[Figure]

**Figure 7.** (a) Summary of total RF, RF for major aerosol components due to aerosol-radiation interactions (RFari), and aerosol effects on clouds (CRF) in East China and the north Pacific due to the changes in anthropogenic emissions between 2008 and 2016; (b) Aerosol DRF and CRF owing to projected anthropogenic emissions for 2030 and 2050 relative to 2016.

---

## Author Response (AR2)

**Response to Referee 1**

Referee 1: The authors have addressed most of my previous comments. Some concerns remain:

1) Regarding my comment on the CRF estimates, there is no detailed description on how the scaling to AR5 forcing is done and, more importantly, what the justification is. I don't think the CRF estimated from the simulations in this study can be scaled to the AR5 forcing for the following reasons:

• The AR5 effective radiative forcing is estimated for the global present-day condition, compared to the preindustrial condition, which is different from the contrast between 2008 and 2016 with emission changes only in East Asia.

• I also had a comment about the difference in methodology for cloud forcing estimate between this study (using the Ghan method) and the AR5. The cloud radiative forcings are incomparable even just for East Asia.

• On the other hand, if the emission reductions in China during 2013-2017 had a significant impact on the estimated CRF over East Asia, which was NOT included in the AR5 forcing estimates, how can the AR5 forcing be relevant to the East Asia CRF in this study?

**Response:** We sincerely appreciate the referee's further comments, as they are helpful for us to improve the manuscript, especially for the description on the calculation of cloud radiative forcings. To address the concerns from the referee, we'd like to show the followings:

1) First, we explain why and how we scaled the aerosol CRF to match the IPCC AR5 estimate.

To compared with the AR5 cloud radiative forcing, we have performed the global simulation using the preindustrial condition (emissions for 1750) and obtained a global and annual mean aerosol CRF (SW+LW) of -2.7 W m-2 between 1750 and 2008, which is much more negative than the best estimate (-0.5 W m-2) as well as the median value of the multiple model results (-1.4 W m-2) in the IPCC AR5. Our result is close to but still more negative than the estimate (-1.7 W m-2) by another CAM5 model in Ghan et al. (2012).

Such gaps should have been mainly originated from the differences between climate models in processes controlling cloud amounts and lifetime and how aerosols affect cloud albedo (Zelinka et al, 2014). We therefore infer the possibly higher CRF by aerosol reductions over East Asia between 2008 and 2016 in our simulations. A constant scaling factor is derived from the ratio of the global mean CRF in our simulations between preindustrial and present-day conditions to the corresponding AR5 estimate. The scaling can yield a reasonable range of CRF estimates in the context with previous CRF estimates.

(2) As the referee commented, the methodology for cloud forcing estimate is different with the IPCC AR5 because this study followed the methodology in Ghan, 2013, which attributed radiation changes (+0.42 W m-2 globally) induced by abovecloud light absorbing aerosols to DRF rather than CRF. Here, if we add this term to our global mean industrial-era CRF estimate, it is changed to -2.3 W m-2 and still more negative than the best estimate in the IPCC AR5.

The best estimate of ERFaci in IPCC AR5 was given by the expert judgement based on existed studies using climate models and observations. The estimation methods and model structures related to clouds and aerosols differed between each other. One of the model estimates was given by Ghan et al. (2012), in which the methodology to calculate aerosol forcings was used in our study. Thus, it is reasonable that we compared the global mean aerosol CRF in this study with the IPCC AR5 estimates.

(3) The global and annual mean CRF for 1750–2008 was used for comparison with the IPCC AR5 result for 1750–2011 to derive the scaling factor. This factor was then used to adjust the CRF estimates for other periods (2008–2016). In another word, the emission changes between 2008 and 2016 cannot influence the scaling factor.

To sum up, we clarify the methodology for CRF estimates (please see Section 2.2) and show the CRF values both with and without the scaling (following the suggestions by another referee) in the revised manuscript. Please see Line 297-311 in Section 3.3.

Referee: 2) The aerosol radiative forcing is largely determined by column AOD and the vertical distributions. Fig. S5 shows that the CAM5 AOD in East Asia and over the North Pacific is lower than both MISR and MODIS observations. The model also has a large bias in simulating the BC profiles (Fig. S1). Please clarify how this would affect the estimated BC direct forcing. How about the sulfate profiles?

**Response:** Accepted. We add more discussions on the potential causes of model underestimation of AOD temporal changes, and suggest that the model treatments of dust aerosols and biomass burning emissions could be partially responsible for the gap between simulations and observations.

The BC profiles in Fig. S1 reflect the model capability in simulating BC emissions and their transport from the East Asia continental sources to the remote atmosphere. In line with previous studies for the same region (see figures in Schwarz et al., 2013), the differences between the simulated and observed BC profiles in the troposphere are generally within one order of magnitude. We point out how the bias in the profiles may affect the BC DRF in the northern Pacific in the manuscript. Please see Line 187-190 in the revised manuscript.

To our knowledge, observations of sulfate profiles for a long period (>1 month) are not available in China and continental outflows. But we show that the simulated

aerosol extinction profiles from the surface to the upper troposphere are in general agreement with corresponding CALIPSO data in China. Please see Fig. S7 and Line 249-263 in the revised manuscript.

Referee: 3) Please include the information of simulations in the caption of Fig. 1. It appears that the surface sulfate and BC concentrations are unchanged between 2008 and 2016 over regions other than East Asia. This can be very misleading. The other regions probably should be masked out if their emissions are fixed at 2008. Emissions in the surrounding regions, especially in South Asia, has been increasing from 2008 to 2016. As also shown in the MISR and MODIS observations (Fig. S4), AOD in South Asia had a significant increase from 2008 to 2016. The non-local emissions can have an impact on aerosols in East Asia. This raises a concern about the estimated forcing in East Asia.

**Response:** Accepted. We supplement the information of simulations used in Figure 1. We also point out the probable changes in aerosol concentrations in South Asia between 2008 and 2016 that were not reproduced in these simulations. Please see revised Figure 1 caption.

This study focuses on how the aerosol forcings respond to changes in anthropogenic emissions from China, and the contribution of foreign emissions on the aerosol concentration changes is suggested to be negligible during the recent decade (Yang et al., 2018). We have described the use of emissions. Please see Section 2.3.

Referee: 4) Line 306-319: As brought up previously, I don't think it's ok to provide the forcing trends based on results from two individual years. The emission reductions started from 2013. There is no physical basis to suggest the linear forcing trends. The estimated trends should be removed or changed to forcing difference instead.

**Response:** Accepted. We change them to forcing differences and reword the related sentences. Please see Line 312-328 in the revised manuscript.

**References:**

Ghan, S. J., 2013: Technical Note: Estimating aerosol effects on cloud radiative forcing. Atmos. Chem. Phys., 13, 9971-9974.

Ghan, S. J., X. Liu, R. C. Easter, R. Zaveri, P. J. Rasch, J. H. Yoon, and B. Eaton, 2012: Toward a Minimal Representation of Aerosols in Climate Models: Comparative Decomposition of Aerosol Direct, Semidirect, and Indirect Radiative Forcing. J. Clim., 25, 6461-6476.

Schwarz, J. P., and Coauthors, 2013: Global-scale seasonally resolved black carbon vertical profiles over the Pacific. Geophys Res Lett, 40, 5542-5547.

Yang, Y., and Coauthors, 2018: Recent intensification of winter haze in China linked to foreign emissions and meteorology. Sci. Rep., 8, 2107.

**Response to Referee 3**

Referee 3: In this work, the authors simulate the radiative effect of the reduction in Chinese aerosol emissions over the 2008-2016 period. They show that the reduction in aerosol emissions produced a significant reduction in the aerosol burden and radiative forcing. The combination of the SO2 and black carbon emissions reductions produces a larger effect that the effects of the individual components due to interactions between them.

The paper demonstrates some interesting effects, particularly the interaction between different aerosol components in an internally mixed aerosol scheme. Unfortunately, some of the method for calculating radiative forcing values (particularly the cloud component) is unclear, which makes interpretation of some of the results difficult.

(I have been asked to comment primarily on the radiative forcing aspects).

**Response:** Thanks very much for the referee's helpful comments. We accepted all of them and revised our manuscript carefully. Please see the point-to-point responses below.

**Major points**

Referee 3: The scaling to the AR5 mean value is unclear - I assume it means that the ERFaci forcing pattern in the model (for the 2008 simulation) is scaled such that the global mean is -0.5 Wm-2. This means that the same scaling factor is applied everywhere? This should be explained clearly as it is important for the interpretation of the results.

**Response:** Accepted. We detail why and how we scaled the cloud radiative forcings (CRFs).

We have performed a global simulation using the preindustrial condition (emissions for 1750) and obtained a global and annual mean aerosol CRF of -2.7 W m-2 between 1750 and 2008. A scaling factor is derived from the ratio of this industrialera CRF simulation result to the corresponding best estimate of ERFaci in IPCC AR5 (-0.5 W m-2). We then used this globally constant factor (5.4) to scale down the online-calculated CRF in China and northern Pacific regions for 2008–2016.

Details for the scaling method and the results with and without the scaling have been added in the revised text. Please see Section 2.2 and Line 297-311 in the revised manuscript.

Referee 3: I am not convinced that this scaling to the AR5 value for the ERFaci is appropriate. The spatial pattern of the ERF and particularly the cloud component varies significantly between models (e.g. Zelinka et al, JGR, 2014) and between the RFaci and adjustments (e.g. Gryspeerdt et al, ACP, 2020). This means that the ratio of the forcing in the study region to the global mean varies between models and processes. The AR5 estimate is based primarily on observation-based estimates, which themselves have been suggested to underestimate the ERFaci in recent reviews (e.g. Bellouin et al, Rev. Geophys, 2020).

**Response**: Accepted. The reasons for the scaling are given in the revised text and mainly include:

1) We find that the global and annual mean industrial-era aerosol CRF ( $-2.7 \text{ W m}^{-2}$ ) calculated online in our model is more negative than the estimates given in the IPCC AR5 and many other reports, like Zelinka et al. (2014) and Bellouin et al. (2020). The magnitude of this estimate is approximately at the upper bound of the intermodel spread in ERFaci. Not only for the global-mean value, the regional estimate over China ( $-20 \text{ to} -10 \text{ W m}^{-2}$ ) is also more negative than previous studies, like Ghan et al. (2012) and Zelinka et al. (2014). So it is plausible that the online-calculated CRF over China for the 2008–2016 period is too high.

2) The large gap is probably caused by the systematic differences between climate models in physical processes controlling cloud amounts and aerosol effects on cloud albedo (Zelinka et al., 2014). The globally constant scaling factor of 5.4 used in this study is based on IPCC AR5 and may vary if we used other model results as reference. Since the IPCC AR5 estimate has been suggested to be underestimated (mentioned by the referee and Gryspeerdt et al, 2020), the CRF calculations for China during 2008–2016 with and without the scaling can be regarded as the lower and upper bounds.

Therefore, we follow the referee's suggestion to show the original global estimate first and then compare it with other reports. Both our results with and without the scaling for East Asia and outflows are shown in the main text. The scaling cannot influence the major conclusion, but importantly gives a reasonable range of CRF estimates. All the references mentioned here are cited in the revised text. Please see Section 2.2 and Line 297-311.

Referee 3: Given it is not clear why the global mean ERFaci is over/underestimated in this work (and it doesn't appear to be stated anywhere), it is not clear that a scaling derived at a global level is applicable to forcings calculated only over China and the north Pacific. Are both 2008 and 2016 simulations scaled to the AR5 value? As mentioned by another reviewer, if the effect of these emission reductions is significant, this creates a further pattern effect which impacts the scaling performed.

**Response:** Accepted. The scaling factor is derived from the comparison of our 1750 and 2008 simulations with the corresponding estimate from the IPCC AR5 (1750–2011). We used this factor to scale down the ERFaci for the period 2008–2016 globally including over China and northern Pacific Ocean. Therefore, the emission reduction between 2008 and 2016 does not influence the scaling factor. Please also refer to the reason for the scaling of ERFaci estimate shown in the last response.

Referee 3: It might be more appropriate to quote the model values (unscaled) and note that the model over/underestimates the global ERFaci. Where the forcing values need to be compared to the AR5 forcings/a measured value, this difference/scaling factor could be noted.

**Response:** Accepted. We point out the differences of global CRF forcing estimates between our model and the IPCC AR5 (please see Section 2.2), and show the estimates of aerosol forcings in China and outflow regions with and without the scaling. Please see Line 297-311.

**Specific points**

L112 - The contribution on meteorology vs aerosol changes is a potentially important one. Given the simulated change in AOD is smaller than observed when using the nudged simulation, is the observed AOD change reproduced when comparing the new 2016 simulations the authors have done?

**Response:** Accepted. The comparison with the observed AOD change is actually not improved in the new 2016 simulation (that is, the contribution of meteorology is not important). To better interpret the bias of AOD changes, we made the following efforts in the revised manuscript.

1) The modeling of sulfate concentrations was improved by comparing to corresponding observations (Fig. 1a). We increased the  $SO_2$  uptake coefficients for the heterogeneous sulfate formation in each simulation. The revised coefficients lie in the range of previous studies (e.g., Huang et al., 2014; Wang et al., 2015). The greater reductions of sulfate concentrations in our simulations improved the comparison with the observed AOD change.

2) We add the comparison of modeled aerosol extinction profiles with the corresponding CALIPSO data in China. The results suggest that the inadequate treatments of biomass burning emissions and dust aerosols in our simulations could be partially responsible for the AOD underestimation.

Please see Figs. S7-S8 and Line 226-266 in the revised manuscript.

**L148 - HIAPER Pole-to-pole**

Response: Accepted. Please see Line 150.

L171 - 'The variation' - in this and other locations, I think 'variation' is used to mean 'change'. Is that the case?

Response: Accepted. We replace 'variation' to 'change' throughout the manuscript.

L230 - This would be a great opportunity to show if the 2016-meteorology simulations match the observations better than the 2008-meteorology simulations

**Response**: Accepted. We find that the 2016-meteorology simulation did not improve the comparison of AOD changes with the observations. The potential factors resulting in the model biases in AOD are discussed in Section 3.2.

**L273 - contribute less**

**Response:** Accepted. We reword it. Please see Line 293.

**L306 - annual-mean**

**Response:** Accepted. We reword it. Please see Line 316.

**References:**

Bellouin, N., and Coauthors, 2020: Bounding Global Aerosol Radiative Forcing of Climate Change. Rev. Geophys., 58, e2019RG000660.

Ghan, S. J., X. Liu, R. C. Easter, R. Zaveri, P. J. Rasch, J. H. Yoon, and B. Eaton, 2012: Toward a Minimal Representation of Aerosols in Climate Models: Comparative Decomposition of Aerosol Direct, Semidirect, and Indirect Radiative Forcing. J. Clim., 25, 6461-6476.

Gryspeerdt, E., and Coauthors, 2020: Surprising similarities in model and observational aerosol radiative forcing estimates. Atmos. Chem. Phys., 20, 613-623.

Zelinka, M. D., T. Andrews, P. M. Forster, and K. E. Taylor, 2014: Quantifying components of aerosol-cloud-radiation interactions in climate models. J. Geophys. Res.-Atmos, 119, 7599-7615.